# Epigenetic profiling of growth plate chondrocytes sheds insight into regulatory genetic variation influencing height

Michael Guo[1,2,3†], Zun Liu[4†], Jessie Willen[4†], Cameron P Shaw[4], Daniel Richard[4,5], Evelyn Jagoda[4], Andrew C Doxey[5], Joel Hirschhorn[1,2,3], Terence D Capellini[1,4*]

[1]Broad Institute of MIT and Harvard, Cambridge, United States; [2]Division of Endocrinology, Boston Children's Hospital, Harvard Medical School, Boston, United States; [3]Department of Genetics, Harvard Medical School, Boston, United States; [4]Department of Human Evolutionary Biology, Harvard University, Cambridge, United States; [5]Department of Biology, University of Waterloo, Waterloo, Canada

**\*For correspondence:** tcapellini@fas.harvard.edu

[†]These authors contributed equally to this work

**Competing interests:** The authors declare that no competing interests exist.

**Abstract** GWAS have identified hundreds of height-associated loci. However, determining causal mechanisms is challenging, especially since height-relevant tissues (e.g. growth plates) are difficult to study. To uncover mechanisms by which height GWAS variants function, we performed epigenetic profiling of murine femoral growth plates. The profiled open chromatin regions recapitulate known chondrocyte and skeletal biology, are enriched at height GWAS loci, particularly near differentially expressed growth plate genes, and enriched for binding motifs of transcription factors with roles in chondrocyte biology. At specific loci, our analyses identified compelling mechanisms for GWAS variants. For example, at *CHSY1*, we identified a candidate causal variant (rs9920291) overlapping an open chromatin region. Reporter assays demonstrated that rs9920291 shows allelic regulatory activity, and CRISPR/Cas9 targeting of human chondrocytes demonstrates that the region regulates *CHSY1* expression. Thus, integrating biologically relevant epigenetic information (here, from growth plates) with genetic association results can identify biological mechanisms important for human growth.

DOI: https://doi.org/10.7554/eLife.29329.001

## Introduction

Human height, a main outcome of skeletal growth, is the product of many biological and environmental interactions spanning numerous cell and tissue types acting during pre- and post-natal development. The growth plate, located at the distal ends of long bones such as the femur, is among the most important tissues influencing height. In the growth plate, condensations of mesenchymal cells differentiate into chondrocytes in the process of endochondral ossification. Chondrocytes in the growth plate reside in longitudinally oriented columns divided in zones (resting, proliferative, pre-hypertrophic, and hypertrophic) each with important functions in allowing the bone to elongate and mature (see *Liu et al., 2017* and *Samsa et al., 2017* for reviews) (*Liu et al., 2017*; *Samsa et al., 2017*). This process of growth plate chondrocyte differentiation and maturation promotes elongation of long bones, which ultimately determines much of human height (*Baron et al., 2015*). These processes are under the influence of many extrinsic and intrinsic signals (*Samsa et al., 2017*; *Lui et al., 2012*; *Kronenberg, 2003*).

Height has been studied for centuries as a model genetic trait, as it is easily measured and highly heritable (typically, 70–90% of variation in height within a population is attributable to genetic

**eLife digest** Humans vary considerably in height, a trait that is partly inherited from each individual's parents. Studies have identified hundreds of small changes in DNA that contribute to differences in human height. These small changes often swap out just one of the four letters that make up the DNA code. Some changes occur within genes, yet most occur in stretches of DNA that do not contain genes. These stretches of DNA likely control whether genes important for the growth of the skeleton are switched on or off.

Cartilage cells, also known as chondrocytes, are important for height. These cells are found near the end of bones in the growth plates, where the bones grow during childhood and adolescence. The on and off switches for growth genes in chondrocytes are unknown. Identifying these genetic switches could help scientists understand how hundreds of small changes in DNA help determine how tall a person will be.

Now, Guo, Liu, Willen et al. identify thousands of switches that turn on and off genes in chondrocytes. In their experiments, chondrocytes were removed from mouse bones and a type of genetic sequencing called ATAC-seq was used to identify the stretches of DNA that act as on/off switches for genes in these cells. Further analysis revealed that many of these same on/off switches occur in human chondrocytes too. Importantly, the experiments showed that many of the small changes in DNA that contribute to differences in human height are also found within these DNA switches.

Guo, Liu, Willen et al. next tested how these switches and height-linked genes interact, and found, for example, that one switch acts to shut off the gene for a protein called chondroitin sulfate synthase 1. People with mutations in this gene can have unusually short stature. However, people with DNA changes in its nearby switch can be taller or shorter than average based on how the DNA influences the switch. More studies might help scientists understand the evolutionary significance of these DNA changes. They will also help determine if the genetic switches in chondrocytes contribute to diseases that affect the bones and joints, like bone cancer or arthritis.

DOI: https://doi.org/10.7554/eLife.29329.002

variation) (*Silventoinen et al., 2003*; *Perola et al., 2007*; *Visscher et al., 2006*; *Visscher et al., 2007*; *Galton, 1886*). Height and body proportions are likely selected and thus are interesting from an evolutionary standpoint (*Turchin et al., 2012*). Studies of the genetics of height can shed insight into the mechanisms of childhood growth and into developmental biology of the skeleton (*Baron et al., 2015*; *Dauber et al., 2014*). Additionally, abnormalities of growth—short stature or overgrowth—are among the most common childhood disorders and their pathophysiology is poorly understood. Thus, genetic studies of height could further our understanding of the genetic basis of skeletal disease, evolution, biology, and normal human growth and development.

Recently, genome-wide association studies (GWAS) have been used to study the genetics of height, as well as many other traits and disorders (*Wood et al., 2014*). This approach systematically tests each common variant in the genome for association with a phenotype of interest, such as adult height, with the goal of identifying relevant biology, either from the identities of the genes near the associated variants, or by deducing mechanism from the associated variants themselves. GWAS for height are among the largest conducted thus far (sample sizes > 250,000) and have identified more associations than any other phenotype, with nearly 700 regions of the genome (or loci) robustly associated with height (*Wood et al., 2014*). The finding of so many loci influencing height, with the number expected to increase with larger sample sizes, supports the assertion that height is highly polygenic. Although each of the variants at these loci have only a small effect on height (typically much less than 0.5 cm per allele) (*Wood et al., 2014*), the genes within the associated loci as a group highlight important mechanisms influencing growth. For example, height variants are enriched for genes implicated in numerous biological processes relevant to growth, including embryonic stem cell function, long bone development, and spine length, among a number of other processes (*Wood et al., 2014*). Height variants are also preferentially located near genes that are differentially expressed in the growth plate (*Lui et al., 2012*).

However, as with all GWAS, moving from genetic associations to biological mechanisms at any individual locus can be challenging. There are two primary reasons for this difficulty. First, there is extensive linkage disequilibrium (LD) in the human genome, meaning that the genotypes at nearby genetic variants are often tightly correlated (*Gabriel et al., 2002*). The consequence of LD is that at any particular locus, there will be many variants that have indistinguishable associations, and determining which variant(s) are in fact causal (rather than associated due to LD with a causal variant) remains difficult. Second, most GWAS variants (>80%) reside in non-coding regions (*Gusev et al., 2014*), with no nearby coding variants that can account for the associations. Non-coding variants are challenging to interpret because there is no universal regulatory code to infer function/mechanism, and these variants can act to influence gene expression at long distances and in highly tissue-specific manners (*Bernstein and ENCODE Project Consortium, 2012*). Various approaches have been pursued for identifying the causal variants and/or mechanisms, including overlapping associated variants with epigenetic datasets in appropriate tissues/cell-types, statistical genetic methods leveraging subtle variations in LD patterns (called 'fine mapping'), and in vitro and in vivo functional assays (*Tak and Farnham, 2015*). It is also becoming increasingly clear that GWAS variants tend to fall in regions that have particular epigenetically defined states (such as enhancers or repressors) (*Finucane et al., 2015*; *Farh et al., 2015*). Thus, the overlap between GWAS and the appropriate epigenetic annotations can help refine the identities of causal variants, helping to decipher mechanism. However, epigenetic data may not exist for the biologically relevant cell type, and there is no universally successful approach to translating non-coding associations from GWAS into biological mechanisms.

Although the large number of associated variants offers advantages in making biological inferences for the set of associated variants as a whole (*Lango Allen et al., 2010*), defining functional mechanisms of individual height GWAS variants faces specific challenges. Height variants likely act in many tissues and stages of development, including not only the growth plate but also embryonic cells, endocrine tissues such as the pituitary, and bone, among others (*Wood et al., 2014*). Human growth is also a whole-organism phenotype and is less amenable to study in cell-intrinsic models, making functional approaches particularly challenging. Finally, skeletal tissues can be highly heterogeneous and challenging to isolate because in their mature form they are surrounded by a hard extracellular matrix and strongly adhering soft tissues such as tendons, ligaments, and muscles. These factors make it difficult to isolate the large numbers of biologically relevant cells that are needed for functional experiments or for traditional epigenomic assays such as ChIP-seq (*Furey, 2012*).

In this study, we profile the epigenetic landscape of mouse growth plate chondrocytes using a new approach called ATAC-seq (*Buenrostro et al., 2015*, *2013*), which allows for accurate profiling of open chromatin regions using relatively few cells (~50,000). We then use the epigenetic landscape in growth plate chondrocytes to gain new insights into the mechanisms of non-coding elements in the genome influencing expression of growth plate genes. Finally, we use these epigenetic marks, in the context of statistical fine-mapping results and gene expression profiles, to better understand height GWAS variants and the mechanisms by which they may function to influence height in humans. These studies illustrate a path that can help decipher mechanism from GWAS, even when the relevant cell type is relatively inaccessible in humans.

## Results

### Generation of mouse femoral growth plate epigenetic profiles

In order to identify open chromatin regions involved in long bone development, we performed ATAC-seq (*Buenrostro et al., 2015*, *2013*) on E15.5 (Theiler Stage 23) mouse proximal and distal femoral growth zones, regions comprised predominately of the growth plate proper, the overlying perichondrium, and the cartilaginous super-structures that dictate adult bony morphology (Materials and methods). This stage of mouse embryogenesis corresponds to approximately E48-51 (Carnegie Stage 19) of human development and reflects a developmental window when the growth plate proper possesses all chondrocyte zones (resting, proliferative, pre-hypertrophic, hypertrophic zone) and is undergoing rapid cell proliferation and differentiation during endochondral ossification. Given that it remains unclear which sub-region(s) (i.e. chondrocyte zones) of the growth plate are

under genetic influence to drive height variation, we performed our experiments on tissues that include all chondrocyte zones (including perichondrium) and cartilaginous tissues, excluding regions indicative of the osteogenic invasion in the developing bone collar (*Figure 1a*). We used ATAC-seq to isolate open chromatin regions from the DNA, which were subsequently subjected to next-generation sequencing (Materials and methods; *Supplementary file 1*). This approach yielded 28,257 proximal femur peaks and 27,958 distal femur peaks, with 23,764 shared peaks, 4493 peaks specific to the proximal femur, and 4194 specific to the distal femur (*Figure 1b*). The proximal and distal peaks span approximately 0.79% and 0.76% of the genome, respectively. We also assessed the genomic distribution of proximal and distal femur ATAC-seq peaks and found a characteristic enrichment near gene transcriptional start sites and more distal intronic and intergenic non-coding sequences (*Figure 1—figure supplement 1*).

Since a main goal was to understand the gene regulatory functions of variants influencing human height, we next mapped the locations of each mouse femur open chromatin region to the human genome (Materials and methods). Our liftover resulted in 24,805 proximal femur peaks and 24,788 distal femur peaks, representing approximately 87.7% and 88.6% of sites identified in mouse, respectively. These sites that were capable of liftover to human likely reflect relatively conserved genes involved in vertebrate musculoskeletal development (*Shubin et al., 1997*; *Petit et al., 2017*). The union of the two sets contained 25,829 peaks (*Figure 1—figure supplement 1*).

To determine whether ATAC-seq peaks are enriched near gene sets/pathways with specific functions, we performed an enrichment analysis using Genomic Regions Enrichment of Annotations Tool (GREAT) (*McLean et al., 2010*), which identifies curated gene sets enriched near epigenetic peaks. Analyses of each proximal, distal, and union femur datasets yielded general GO Biological Process and GO Molecular Processes terms related to basic cell biological processes, as well as phenotypes related to a number of biological tissues including skeletal tissues (*Supplementary file 2–4*). They also yielded phenotype terms related to general skeletal development, chondrogenesis, and long bone skeletal phenotypes, such as 'Metaphyseal widening', 'Abnormality of the femoral epiphyses', and 'Abnormality of the femoral head' (*Figure 1c*). Thus, ATAC-seq datasets generated on each specific femoral growth plate in mice reflect in part highly relevant anatomy, as well as general aspects of skeletal development in humans (see Discussion).

The ATAC-seq peaks we identified also highlight biological processes that are important in the growth plate. For example, GREAT analyses uncovered enrichment for several relevant GO Biological Processes such as 'Cartilage condensation', 'Positive regulation of chondrocyte differentiation', and 'Chondrocyte proliferation' (*Supplementary file 2–4*). Additionally, we illustrate ATAC-seq peaks at several well-known growth plate genes including *Col2a1*, *Fgfr1*, and *Pth1r* (*Figure 1—figure supplement 2*) (*Long and Ornitz, 2013*).

## ATAC-seq peaks overlap other skeletal and chondrocyte epigenetic data sets

We next wanted to validate that our ATAC-seq datasets reflect open chromatin regions important to mouse chondrocyte biology. To do so, we analyzed a previous dataset that used ChIP-seq to profile Sox9 binding in mouse rib chondrocytes (*Ohba et al., 2015*). We found extensive overlap between Sox9-occupied sites and our open chromatin ATAC-seq peaks. For example, 86.4% of Class I Sox9 elements (i.e. elements clustering near transcriptional start sites of highly expressed non-chondrocyte-specific genes), 44.6% of Class II Sox9 elements (i.e. elements directing chondrocyte-related gene activity through direct Sox9 dimer binding), and 93.8% of Sox9 super-enhancer regions (i.e. enhancer complexes involved in chondrocyte cell identity) overlap with our femur ATAC-seq peaks (p<0.001 for all three sets, Materials and methods, *Figure 2a*).

We also examined overlap between our dataset and various histone modifications in the mouse rib chondrocytes (Materials and methods) (*Figure 2b*) (*Ohba et al., 2015*). We found that 85.1% of RNA Pol II transcriptionally active promoters, 56.9% of H3K4me2 active promoter sites, 86.7% of H3K4me3 active promoter sites, 87.4% of H3K27ac active enhancer sites, and 59.4% of p300 histone acetyltransferase active enhancer sites overlapped with femur ATAC-seq peaks. Conversely, our ATAC-seq datasets displayed relatively lower overlap with markers of gene repression; that is, only 33.7% of H3K27me3-repressed enhancer sites overlap with femur ATAC-seq signals (*Figure 2b*). Based on permutation analyses, the enrichments of all these histone modification sets were

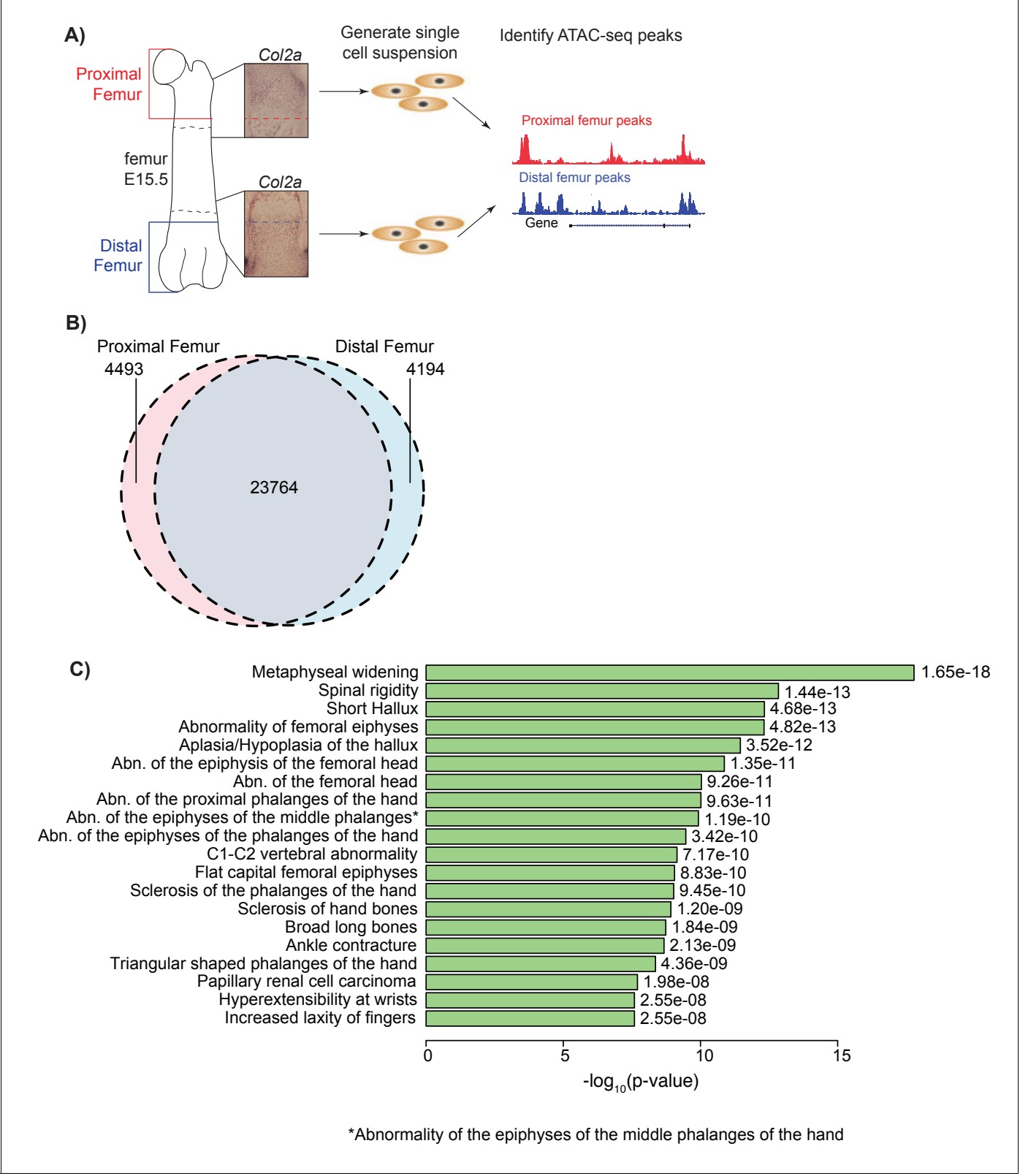

**Figure 1.** ATAC-seq on in vivo collected mouse femoral growth plate zones. (**A**) Pipeline of ATAC-seq on the embryonic mouse femur. Proximal (red) and distal (blue) samples were extracted from E15.5 femoral growth plates, as labeled by *Col2a1* expression (insets). Tissues were harvested just proximal (red dotted lines in inset) or distal (blue dotted lines in inset) to the forming bone collar (outlined by dashed black lines). Cells were then treated to generate a single-cell suspension. Approximately 50,000 cells per sample were utilized for subsequent ATAC-seq. Open chromatin regions

*Figure 1 continued on next page*

*Figure 1 continued*

were then identified from the ATAC-seq data. ATAC-seq protocol adapted from *Buenrostro et al. (2013)*. (B) Venn diagram shows the number of ATAC-seq peaks in the proximal and distal samples. (C) ATAC-seq peaks (after subtracting peaks shared in brain ATAC-seq) were analyzed for enrichment in gene sets using GREAT. Horizontal bars show enrichment p values (shown as –log₁₀(p value)) for Human Phenotypes. The union of proximal and distal femur peaks was used.

DOI: https://doi.org/10.7554/eLife.29329.003

The following figure supplements are available for figure 1:

**Figure supplement 1.** Details on chondrocyte ATAC-seq regions.

DOI: https://doi.org/10.7554/eLife.29329.004

**Figure supplement 2.** ATAC-seq peaks at known chondrocyte loci *Col2a1* (A), *Fgfr1* (B), and *Pth1r* (C).

DOI: https://doi.org/10.7554/eLife.29329.005

significant at p<0.001. Thus, we found that femoral ATAC-seq peaks show marked evidence of active chromatin state and/or gene transcription.

We also wanted to determine if our ATAC-seq peaks reflect regulatory elements involved in human limb development and chondrocyte biology. First, we used a ChIP-seq dataset comprised of H3K27ac peaks found in E47 human limb buds (equivalent to E13.5 mouse limb buds) that are shared with mouse limb buds (*Cotney et al., 2013*). We found that 45.2% of these E47 human limb bud H3K27ac peaks overlapped with ATAC-seq peaks, indicating that many of our mouse femur ATAC-seq peaks reflect active human limb enhancers involved in chondrogenesis (p<0.001, *Figure 2c*). Second, we used an H3K27ac ChIP-seq dataset generated on human adult cultured bone marrow mesenchymal stem-cell-derived chondrocytes (BMDCs) (*Herlofsen et al., 2013*; *Kundaje et al., 2015*) and found that 13.3% of human adult BMDC H3K27ac peaks overlapped our mouse distal femur dataset (p<0.001) (*Figure 2c*), which indicates that our femur ATAC-seq dataset also captures cell-intrinsic properties of chondrogenesis. However, the greater overlap seen in the limb bud dataset as compared to the BMDCs suggests that our dataset may be capturing epigenetic properties present in developing in vivo tissues that are poorly modeled by cell culture systems.

Our ATAC-seq data were also able to identify 14 of 26 previously-discovered active regulatory elements residing near key cartilage genes detected at different time-points and using different assays (e.g. see *Figure 2d*, *Supplementary file 5*). This suggests that mouse femoral ATAC-seq regions have in vivo enhancer activity as determined in independent assays. Overall, these data indicate that ATAC-seq datasets generated on the mouse femoral growth plates match previously expected patterns of regulatory usage important to chondrocyte biology.

## Human height variants are enriched in femoral open chromatin regions

We next sought to use our ATAC-seq dataset to gain insight into height-associated variants identified by GWAS. To do so, we intersected our orthologous human ATAC-seq peaks with 688 height GWAS loci from *Wood et al. (2014)* (nine loci were excluded; see Materials and methods) (*Wood et al., 2014*). Since we do not know which variants are causal and which are associated with height due to LD, for each locus, we identified potentially causal 'proxy' SNPs: those with a correlation of $r^2$ >0.5 to the lead SNP based on the European subset of 1000 Genomes Phase 3 (*Auton et al., 2015*). The number of proxy SNPs ranged from 1 to 3548 for each of these 688 loci (*Figure 3—figure supplement 1*).

Across the 688 height loci, we found that 317 loci (or ~46%) contained at least one variant that overlapped with an ATAC-seq peak. To test the significance of the observed overlap, we applied GoShifter (see Materials and methods), which performs a shuffling-based approach to evaluate the overlap between an epigenetic data set and GWAS data. We found strong enrichments for the ATAC-seq peak set and height GWAS data (p=0.0059) (*Figure 3—figure supplement 1a*). We repeated this analysis using the Sox9 binding and histone modifications in mouse rib chondrocytes, human BMDCs, and mouse embryonic limb datasets from above, as well as DNaseI hypersensitivity sites from mouse brain, liver, and lung (*Bernstein and ENCODE Project Consortium, 2012*; *Ohba et al., 2015*; *Cotney et al., 2013*; *Yue et al., 2014*). Only the mouse rib chondrocyte H3K4me2 peaks reached statistical significance (p=0.0237), and was far less significant than our ATAC-seq data. This suggests that our ATAC-seq data captures functional genetic variation

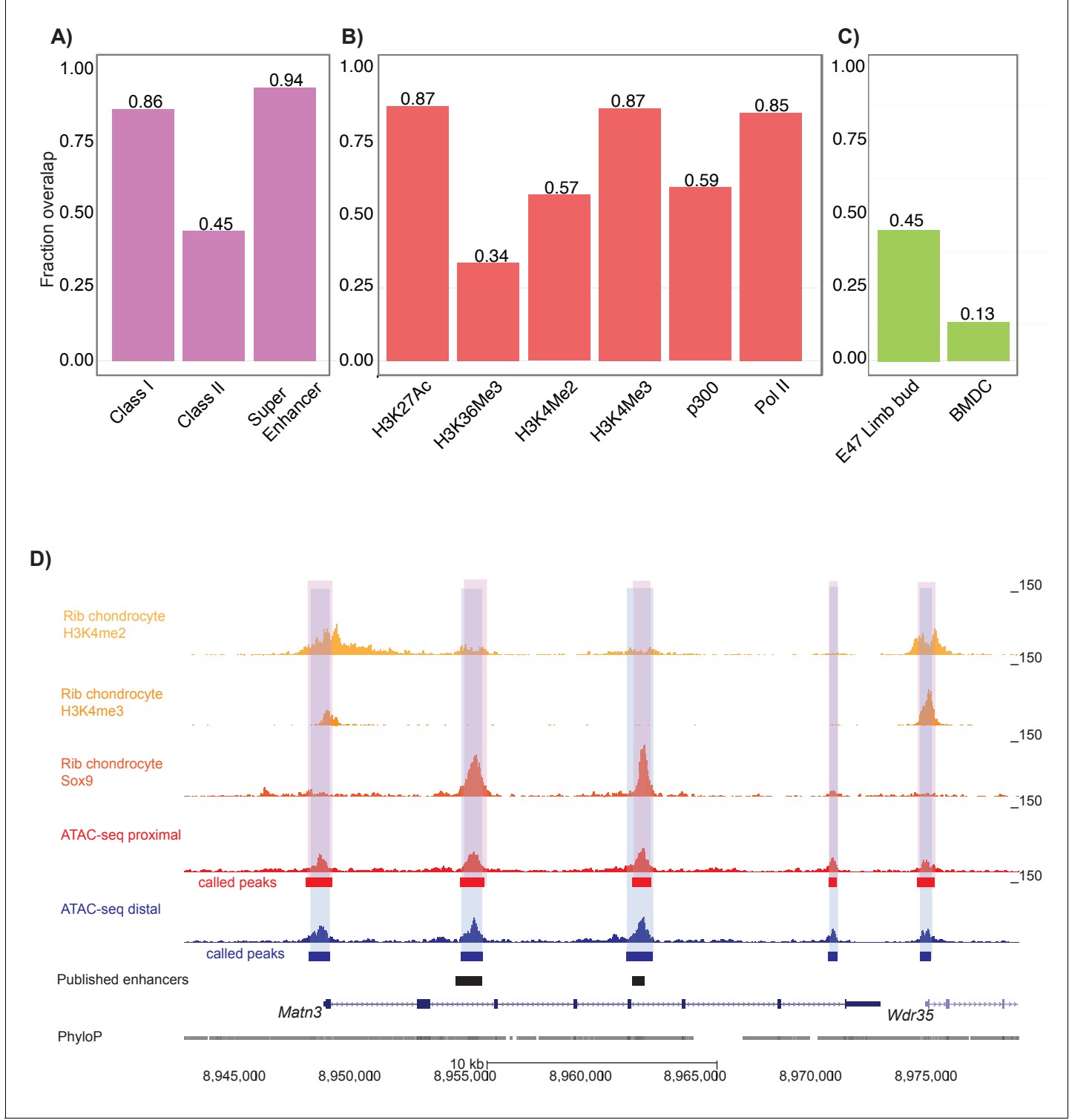

**Figure 2.** Mouse proximal and distal ATAC-seq datasets capture chondrocyte biology in mouse and humans. (A) Proportion of mouse Sox9 Class I, Class II, and super enhancer sites (*Ohba et al., 2015*) overlapping with ATAC-seq peaks. (B) Proportion of mouse rib chondrocyte H3K27ac, H3K36me3, H3K4me2, H3K4me3, p300, and RNA Pol II sites overlapping with ATAC-seq peaks. (C) Proportion of human embryonic limb E47 H3K27ac (*Cotney et al., 2013*) and human BMDC H3K27ac sites (*Kundaje et al., 2015*) overlapping with ATAC-seq peaks. (D) ATAC-seq peaks at the *Matn3* locus. Published rib chondrocyte peaks for H3K4me2, H3K4me3, and Sox9 are shown in shades of orange. Proximal ATAC-seq peaks are shown in red, with called peaks shown as bars below. Distal ATAC-seq peaks are shown in blue, with called peaks shown as bars below. The locations of previously published chondrocyte enhancers are shown as black bars.

DOI: https://doi.org/10.7554/eLife.29329.006

influencing height that is not captured by the other datasets. Also, as no enrichment was observed for other mouse tissues from ENCODE, this suggests that the observed enrichment is unlikely due to artifacts stemming from our use of mouse tissues (e.g. from conserved regions being lifted over).

To further demonstrate the specificity of our ATAC-seq peaks for capturing height GWAS signals, we repeated the GoShifter analysis by examining the enrichment of GWAS results for other complex traits/diseases at our ATAC-seq peaks. These other traits/diseases were chosen as they have no obvious connection to height and included age of menarche (*Day et al., 2017*), body mass index (*Locke et al., 2015*), coronary artery disease risk (*Schunkert et al., 2011*), LDL cholesterol (*Teslovich et al., 2010*), mean corpuscular volume (*van der Harst et al., 2012*), schizophrenia risk (*Ripke and Schizophrenia Working Group of the Psychiatric Genomics Consortium, 2014*), and type 2 diabetes risk (*Morris et al., 2012*). These additional traits/diseases all demonstrated little to no enrichment at ATAC-seq peaks (*Figure 3—figure supplement 1*), suggesting that the growth plate ATAC-seq peaks are specifically capturing height/growth biology.

As a second approach to evaluate overlap between our ATAC-seq peak set and height GWAS data, we generated 1000 sets of random loci which were matched to the actual height GWAS loci based on factors including SNP density, minor allele frequency, number of proxy SNPs in the locus, and gene density (Materials and methods) (*Pers et al., 2015*). For each of the 1000 matched sets of random loci, we repeated the overlap with the ATAC-seq peaks. As none of the matched sets had as much overlap as our height GWAS loci, the enrichment we observed for the height GWAS data is highly significant (p<0.001; z-score: 11.88) (*Figure 3c*).

Notably, we observed that the actual number of SNPs that overlap an ATAC-seq peak for each set was higher than the number of represented loci. In total, there were 928 variants that overlapped with an ATAC-seq peak; these 928 variants were distributed across 317 loci (*Supplementary file 6*; *Figure 3—figure supplement 1*). This finding of more than one overlapping variant per locus may indicate multiple causal variants at each locus, as has been shown for other traits (*Roman et al., 2015*).

## Fine-mapped height variants overlap with femoral growth plate open chromatin regions

An alternative method to narrow down potentially causal signals within GWAS loci is to use statistical fine-mapping, which leverages the association statistics and patterns of LD at each locus to identify which variants at a given locus are most likely to be causal. We applied PICS (*Farh et al., 2015*) to perform statistical fine-mapping of the (*Wood et al., 2014*) height GWAS results (Materials and methods). For each of 688 height-associated loci, we calculated credible set probabilities for all SNPs with an $r^2$ >0.5 to the lead variant based on the European subset of 1000 Genomes Phase 3 (*Auton et al., 2015*). For each locus, we then determined the 95% credible set, which represents the minimum set of SNPs at that locus such that the truly causal SNP has at least a 95% probability of being in the set. Across 688 loci, this yielded a 95% credible set of 32,846 SNPs, which represents approximately a 48.2% reduction from the total number of variants in the GWAS loci. Intersecting these credible set SNPs with femur open chromatin regions revealed 468 overlapping variants spanning 192 of the 688 loci (*Figure 4a*).

## A subset of human height variants in femoral open chromatin regions reside near genes differentially expressed in the growth plate

A previous study profiled expression of mouse growth plate genes and identified a set of 427 genes that were differentially expressed in the growth plate (*Lui et al., 2012*). That study also revealed that height GWAS loci (using a previous smaller height GWAS) are enriched near differentially expressed growth plate genes (*Lui et al., 2012*; *Lango Allen et al., 2010*). We expanded on those results using the most recent height GWAS from *Wood et al. (2014)* and indeed found that height GWAS loci are enriched near 427 differentially expressed growth plate genes (within 100 kb) when compared to the 1000 matched random GWAS loci (p<0.001, z-score: 8.54, *Figure 5a*).

Since we detected enrichment of height GWAS variants near femoral open chromatin regions (see *Figure 3*) and reasoned that many of the variants likely function by influencing expression of growth plate genes, we next identified the number of GWAS variants that fit the criteria of residing in an open chromatin region near a differentially expressed gene (within 100 kb). We again used the

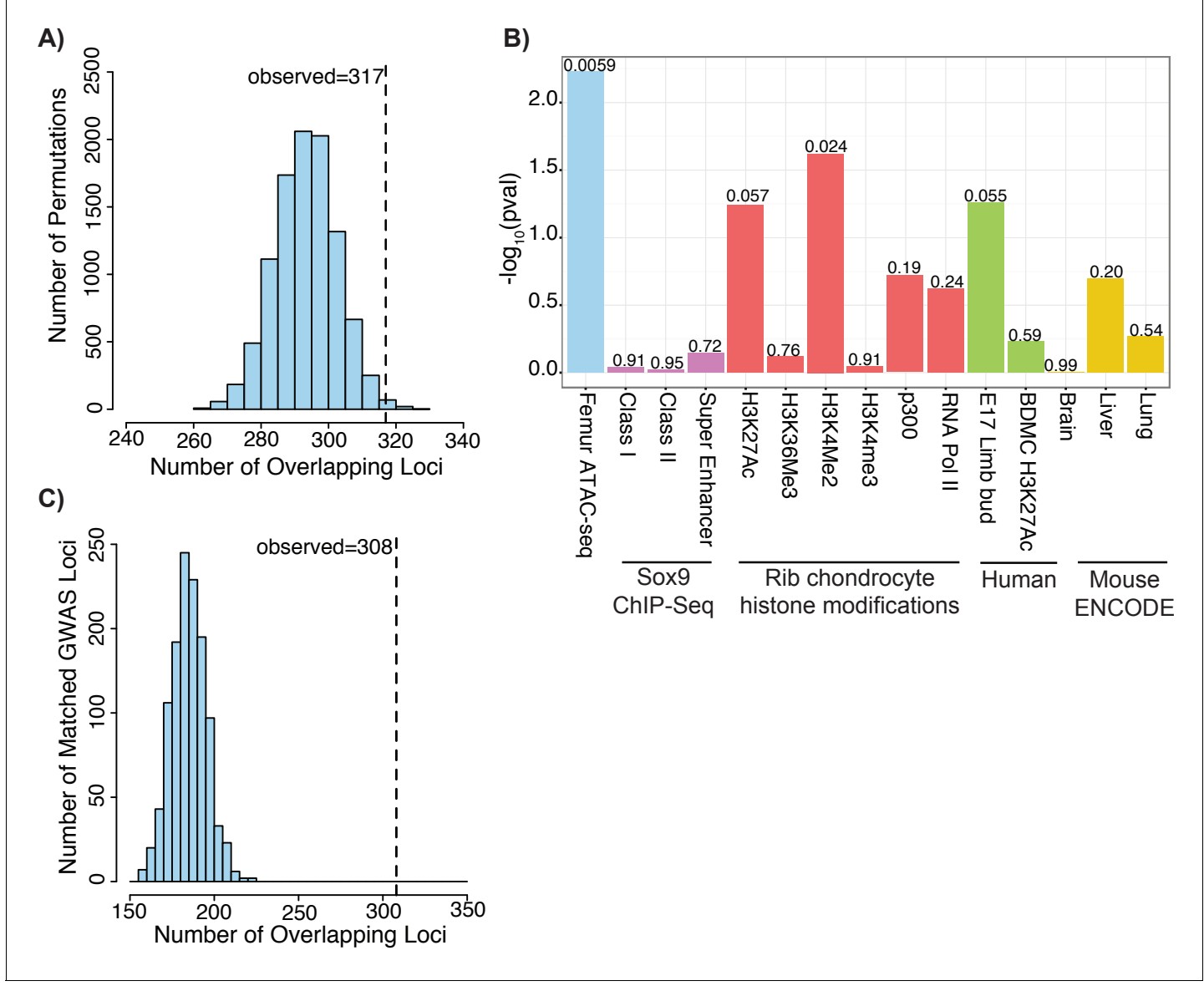

**Figure 3.** Height GWAS loci are enriched in ATAC-seq peaks. (**A**) Enrichment of height GWAS loci in ATAC-seq peaks using GoShifter. Histogram (light blue bars) shows the number of overlaps observed in the 100,000 GoShifter permutations. Dotted line shows the number of overlaps observed in the ATAC-seq data. (**B**) GoShifter analysis p values for the ATAC-seq data, mouse rib chondrocyte Sox9 binding and histone modifications, human embryonic limb E47 H3K27ac, H3K27Ac of BMDCs, and brain, liver, and lung DNaseI hypersensitivity site samples from mouse ENCODE (**C**) Enrichment of height GWAS loci in ATAC-seq peaks using random matched GWAS loci. Histogram (light blue bars) shows the number of overlaps observed in 1000 matched random loci. Dotted line shows the number of overlaps observed in the ATAC-seq data.

DOI: https://doi.org/10.7554/eLife.29329.007

The following figure supplement is available for figure 3:

**Figure supplement 1.** Intersection of GWAS loci and growth plate chondrocyte ATAC-seq.

DOI: https://doi.org/10.7554/eLife.29329.008

688 GWAS loci from *Wood et al. (2014)* and their proxy SNPs ($r^2$ >0.5) and found that 46/688 loci (or ~6.7%) contained at least one GWAS variant that overlapped with an ATAC-seq peak and resided near a differentially expressed gene (*Supplementary file 6*). Using the same 1000 sets of matched random loci as described above, we found that these overlaps were highly significant (p<0.001, z-score: 6.36, *Figure 5b*). These loci likely represent combinations of functional variants

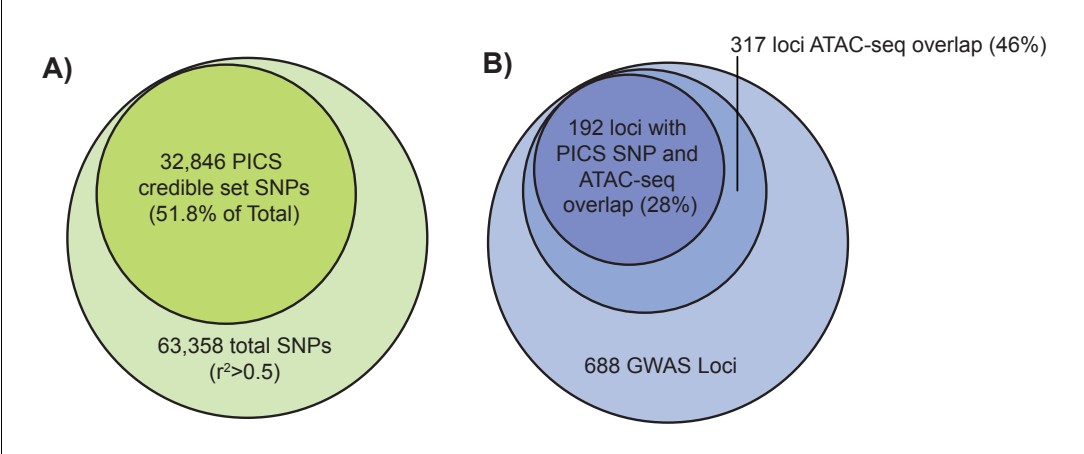

**Figure 4.** Fine-mapping of GWAS height variants. (**left**): Larger light green circle represents all height GWAS SNPs and their proxies ($r^2 > 0.5$). Smaller green circle represents the proportion of these SNPs in the PICS 95% credible set. (**right**) Larger light blue circle represents the 688 height GWAS loci. Medium blue circle represents 317 height GWAS loci overlapping an ATAC-seq peak. Smaller dark blue circle represents 192 loci where a PICS 95% credible set variant overlaps an ATAC-seq peak.
DOI: https://doi.org/10.7554/eLife.29329.009

that overlap an epigenomic element in growth plates that then modulates expression of nearby genes influencing height.

Wood et al., had previously observed that height GWAS SNPs that represent cis eQTLs in whole blood are enriched for expression in cartilage (*Wood et al., 2014*). Here, we sought to evaluate whether differentially expressed growth plate genes that are also eQTLs genes in whole blood are enriched at ATAC-seq peaks. Among the 427 differentially expressed growth plate genes, 157 had

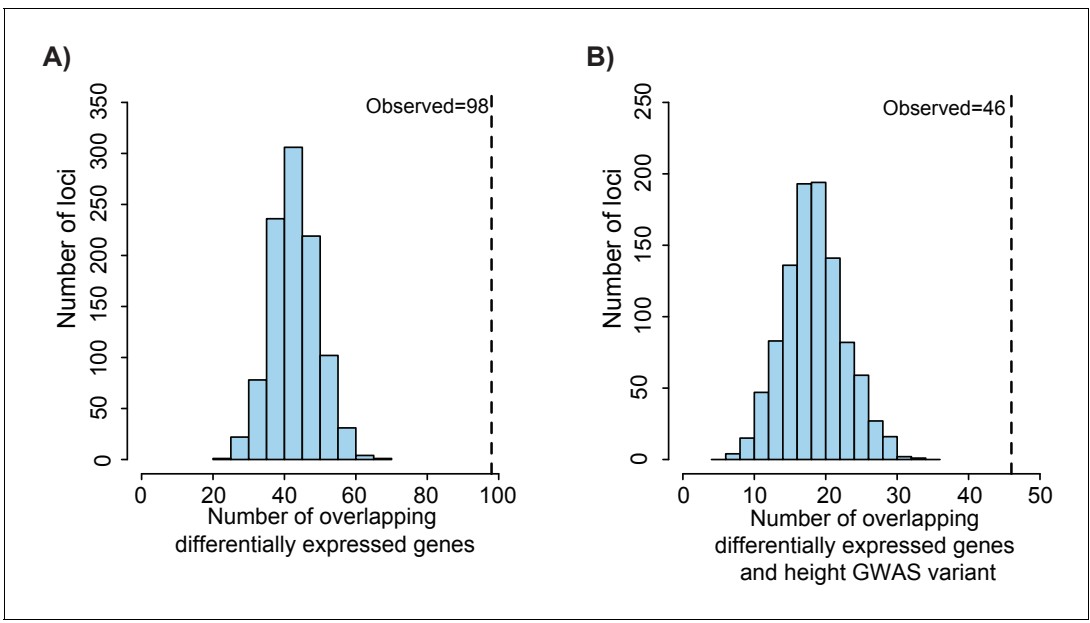

**Figure 5.** Height GWAS loci are enriched in differentially expressed growth plate genes. (**A**) Enrichment of height GWAS loci in differentially expressed growth plate genes. Dotted line shows number of GWAS loci with a nearby differentially expressed growth plate gene (±100 kb). Histogram (light blue bars) shows the number of overlaps observed in 1000 matched random loci. (**B**) Enrichment of height GWAS loci in differentially expressed growth plate genes and ATAC-seq peaks. Same as (**A**), except showing overlap for GWAS loci overlapping differentially expressed growth plate gene (±100 kb) and ATAC-seq peak.
DOI: https://doi.org/10.7554/eLife.29329.010

previously been identified as eQTL genes in whole blood (*Westra et al., 2013*). Using the GoShifter permutation analysis as described above, we demonstrated that the corresponding eQTL variants for these 157 genes are significantly enriched at ATAC-seq peaks (p=0.0004)

Of the 46 loci in the ATAC-seq femur comparison that fall near genes differentially expressed in growth plates, 44 different genes have been identified as possibly being modulated by these putative regulatory variants (note, some genomic loci have multiple genes and/or GWAS loci) (*Supplementary file 6*). These genes represent major signaling molecules (e.g. *IHH*, *HHIP*), transcription factors (e.g. *RUNX2*), extracellular matrix factors (e.g. *CHSY1*, *EXTL1*), among other factors involved in the growth plate in both humans and mice.

## Known and novel targets of human height variants in putative chondrocyte regulatory regions

We next examined some of the most compelling loci from a genetic and mechanistic perspective. We filtered for loci where a height GWAS variant in the PICS 95% credible set overlapped an ATAC-seq peak, and was near a differentially expressed growth plate gene (±100 kb). In total, 59 variants distributed across 26 loci met these criteria. These 59 variants thus have strong evidence for being potentially causal variants and might be prioritized for downstream study (*Supplementary file 6*).

For example, at the *CHSY1* locus, four variants (rs8042551, rs11639408, rs9920291, and rs3911964) met these criteria. Disruption of *Chsy1* in mice and humans leads to severe skeletal phenotypes including long bone length reductions (*Wilson et al., 2012*; *Temtamy et al., 1998*; *Tian et al., 2010*; *Li et al., 2010*). Of the four variants, rs9920291 is predicted to alter HOXD13 binding, as suggested by data generated using universal protein binding microarray technology that demonstrates that the eight base-pair sequence matching the immediate rs9920291 locus disrupts in vitro binding by HOXD13 (*Figure 6a*, *Supplementary file 7*), a homeodomain transcription factor that is expressed in the growth plate (*Kuss et al., 2014*; *Reno et al., 2016*) and has known activating and repressive roles in skeletal development (*Johnson and Tabin, 1997*). We performed a series of functional assays to test the effects of allelic variation at rs9920921. First, using a luciferase reporter assay in T/C-28a2 human chondrocytes, we found that the open chromatin region surrounding rs9920291 acts as a repressor of expression (p=$7.83 \times 10^{-6}$ and $3.6 \times 10^{-4}$ for variants carrying the reference and alternate alleles respectively) (*Figure 6b*). In addition, the alternate T allele, (i.e. the height-increasing allele of rs9920291) makes the regulatory element act as a weaker repressor (p=0.0014). This is consistent with data from GTEx (Lonsdale J, GTEx Consortium, 2013), where rs9920291 acts as an eQTL for *CHSY1* in transformed fibroblasts (p=$1.0 \times 10^{-7}$; *Figure 6—figure supplement 1*), with the T allele correlating with increased *CHSY1* expression levels. Additionally, we leveraged the fact that in HEK-293FT cells, rs9920291 is heterozygous, along with several additional SNPs in the coding region of *CHSY1*. Assaying a heterozygous exonic coding SNP (rs28364839) and a 3'UTR SNP (rs11433), we demonstrate that rs9920291 displays allelic skew in expression (p<0.001 for both assayed variants; *Figure 6d*).

To further demonstrate that the region surrounding rs9920291 displays regulatory activity for *CHSY1*, we performed CRISPR-Cas9 targeting of a 135 bp sequence spanning a conserved portion of the regulatory element containing rs9920291 (guides sg1 and sg4), or an 17 bp sequence immediately surrounding rs9920291 (guides sg6 and sg8). Either deletion led to a significant upregulation of *CHSY1* expression in vitro in T/C-28a2 cells (p=$7.83 \times 10^{-6}$ for the 135 bp deletion and p=$3.64 \times 10^{-4}$ for the 17 bp deletion; *Figure 6d*), confirming the regulatory element's repressor activity. Consistent with our computational predictions that rs9920291 perturbs HOXD13 binding, we found that overexpression of HOXD13 in T/C-28a2 cells significantly upregulated *CHSY1* expression (p=$1.63 \times 10^{-5}$; *Figure 6e*,*Figure 6—figure supplement 1*).

We also highlight a previously identified height-associated variant (rs4911178) at the *GDF5* locus, which resides in an ATAC-seq peak active in femoral growth plates (*Figure 6—figure supplement 2*). We have shown in separate work that the underlying enhancer drives *GDF5* expression in growth plates and that rs4911178 has allele-specific activity(*Capellini et al., 2017*). Thus, our approach of integrating genetic and growth plate epigenetic data has the ability to identify previously discovered cis-regulatory targets that mediate human height variation.

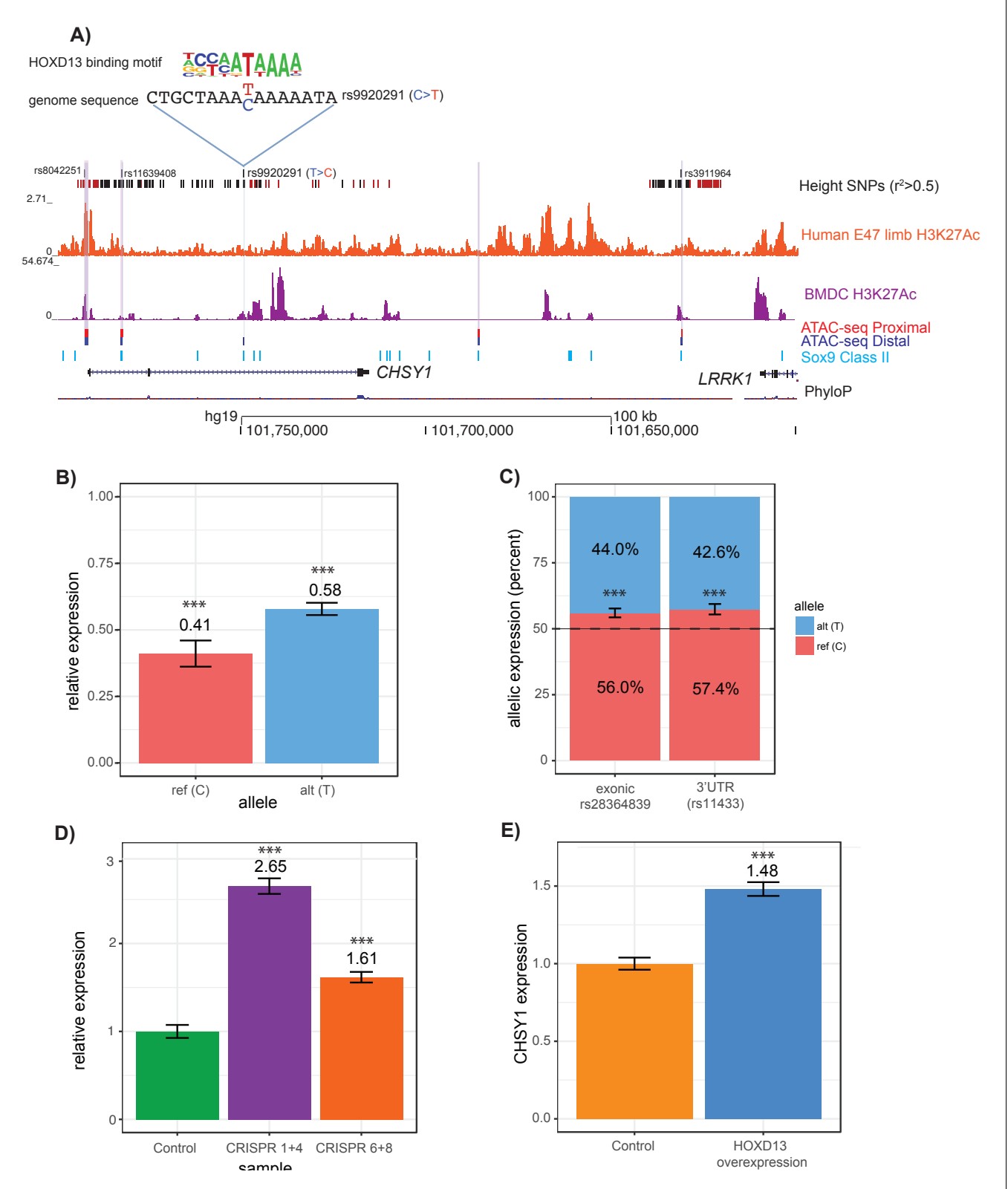

**Figure 6.** Intersections of GWAS height variants with ATAC-seq data identify novel putatively causal variants. (**A**) ATAC-seq height GWAS variant intersections at the *CHSY1* locus refine height GWAS loci to fewer putatively causal variants. Height GWAS SNPs ($r^2$ >0.5 to lead SNP) are shown as black bars, and SNPs in the PICS 95% credible set are shown as red bars. Published H3K27ac ChIP-seq in E47 human limb buds and BMDCs are shown in orange and purple respectively. ATAC-seq peaks lifted over to human are shown for proximal femur (red) and distal femur (blue). Published Sox9

*Figure 6 continued*

class II sites from mouse rib chondrocytes lifted over to human are shown in light blue. Four height SNPs overlap an ATAC-seq peaks, representing putatively causal variants. Of these four variants, rs9920291, a C to T substitution, significantly alters the predicted binding of HOXD13 (UniProbe: HOXD13 C allele z-score = 0.7825; HOXD13 T allele z-score = 5.1269 (also *Supplementary file 7*) (B) Results of a luciferase reporter assay. A 341 bp region surrounding rs9920291 and containing either the reference or alternate allele at rs9920291 was cloned into a luciferase reporter vector and transfected into the human chondrocyte cell line T/C-28a2. Expression was normalized (to 1.0) by luciferase activity of the empty vector luciferase construct. (C) Allelic skew of two *CHSY1* genic variants (rs28364839 which is exonic and rs11433 which is in the 3'UTR) in HEK-293FT cells. Notably, HEK-293FT is heterozygous for rs9920291, along with both assayed variants. (D) Expression of *CHSY1* following CRISPR-Cas9-mediated deletion of regions surrounding rs9920291 (see *Figure 6—figure supplement 1*) in T/C-28a2 chondrocyte cell lines (N = 3 biological replicates). *CHSY1* expression following deletion of an approximate 135 bp region using guides 1 and 4 is shown in purple and expression following deletion of an approximate 17 bp region using guides 6 and 8 is shown in orange. Expression values are normalized to a control CRISPR-Cas9 vector (green). (E) *CHSY1* expression following HOXD13 overexpression in T/C-28a2 cells (N = 3 biological replicates) (see also *Figure 6—figure supplement 1*). Expression is normalized by an empty overexpression vector. ***p<0.001 by unpaired t-test.

DOI: https://doi.org/10.7554/eLife.29329.011

The following figure supplements are available for figure 6:

**Figure supplement 1.** CRISPR-Cas9 targeting of *CHSY1* and HOXD13 overexpression studies.

DOI: https://doi.org/10.7554/eLife.29329.012

**Figure supplement 2.** ATAC-seq height GWAS variant intersections at the *GDF5* locus.

DOI: https://doi.org/10.7554/eLife.29329.013

## Motif analysis of height variants that reside within distal femur ATAC-seq peaks

To expand upon our understanding of transcription factor binding at ATAC-seq open chromatin regions, we performed a series of motif enrichment analyses using HOMER (see Materials and methods). First, we performed a de novo motif analysis on the union of distal and proximal femoral ATAC-seq peaks in mouse to identify enriched transcription-factor-binding motifs. We identified a number of highly enriched motifs relevant to chondrogenesis and other skeletal developmental processes, including *Atf3* (*James et al., 2006*), *Msx2* (*Amano et al., 2008*), *Pbx1* (*Selleri et al., 2001*), and *Ctcf* (*Jerković et al., 2017*) (*Figure 7a*, *Supplementary file 8*, sheet 1). We performed a similar analysis on the liftover human ATAC-seq regions and found enrichments for a similar set of motifs (*Figure 7a*, *Supplementary file 8*, sheet 2).

We next characterized the effect of height GWAS variants that intersect with these motifs within ATAC-seq peaks. Further characterization of the GWAS variants within CTCF motifs indicated a tendency toward disruptive effects (*Figure 7b*). Comparing the height GWAS variants intersecting ATAC-seq peaks to 500 sets of random GWAS variants revealed an enrichment for variants that are predicted to impact CTCF-binding (p<0.01, z-score: 6.01, *Figure 7c*) (see Materials and methods). An example of a variant within a CTCF motif is shown in *Figure 7d*. Additionally, we detected a strong enrichment of height GWAS variants overlapping predicted PBX1 motifs within ATAC-seq regions, when compared to height GWAS variants across the genome (adjusted p value=$1.91 \times 10^{-108}$, *Supplementary file 9*; see Materials and methods).

## Discussion

Like many anthropometric traits, height is highly heritable and influenced by a large number of distinct genetic inputs that influence the development and/or growth of a number of tissues (*Visscher et al., 2010*). Of these tissues, the growth plate plays a key role in determining overall height. Despite decades of research on chondrocyte growth plate biology, we have been limited in our understanding of how height GWAS variants—most of which are non-coding—act in the growth plate to influence height. This is due in part to the paucity of epigenetic data related to the regulatory landscape of the growth plate, as these datasets have been technically challenging to generate. Here, we leveraged a new method (ATAC-seq) (*Buenrostro et al., 2015*, *2013*) to overcome previous technical challenges and reveal the open chromatin regulatory landscape of growth plate chondrocytes. We then analyzed height GWAS data in the context of the femur growth plate epigenetic landscape to gain insight into how height GWAS variants act at the growth plate.

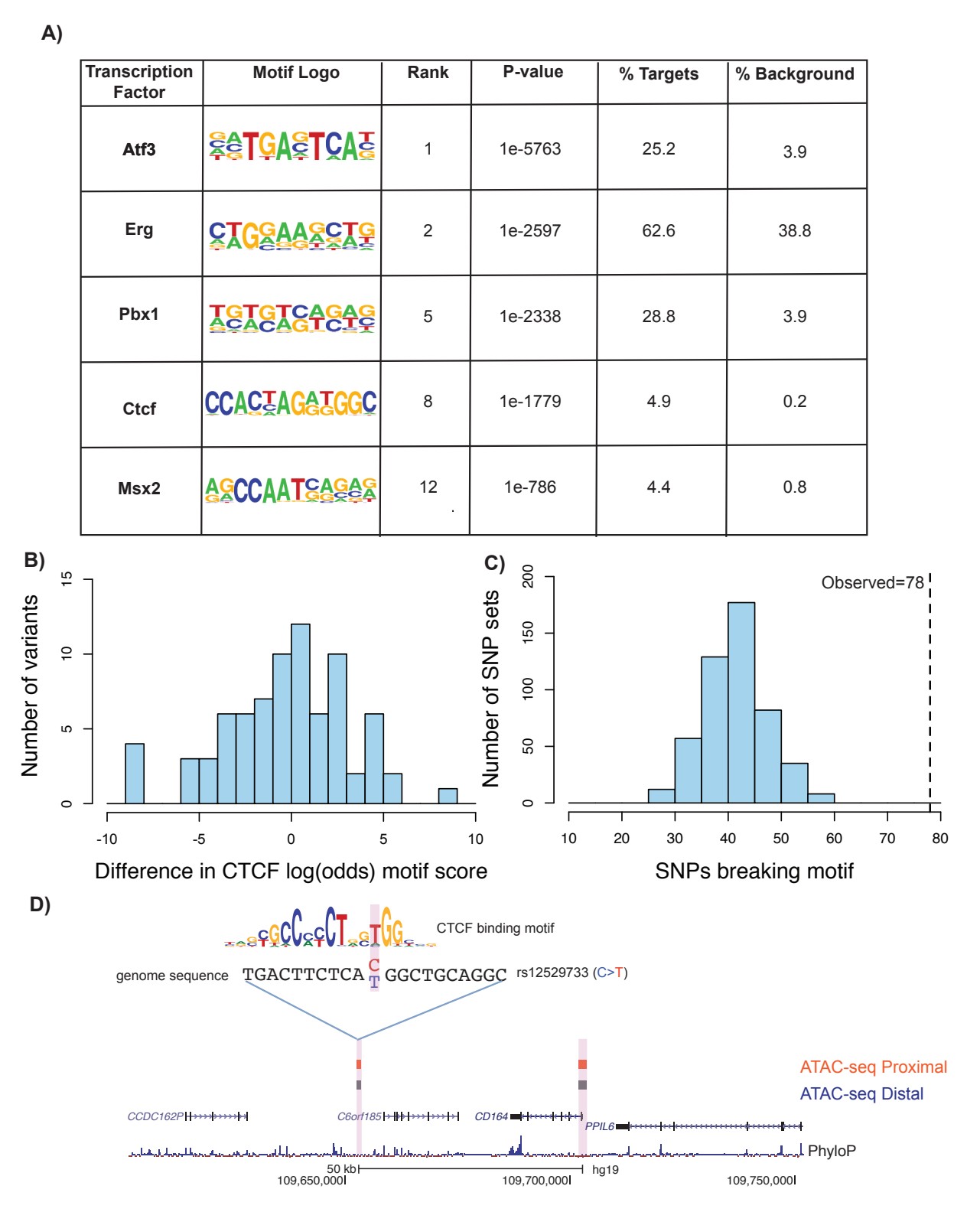

**Figure 7.** Motif disruption by height GWAS variants intersecting ATAC-seq peaks. (**A**) Selected enriched motifs within mouse femoral ATAC-seq open chromatin regions. These motifs were also found following liftover of the peaks to the human genome (see *Supplementary file 8*, sheets 1 and 2). (**B**) Histogram showing distribution of log-odds motif scores of CTCF-disruption for height GWAS variants within ATAC-seq regions. Negative values indicate a disruptive effect of the variant while positive values indicate a 'conservation' of the motif hit by the variant. (**C**) Histogram of number of

*Figure 7 continued on next page*

*Figure 7 continued*

variants predicted to disrupt CTCF motifs within random subsamples of variants. Dotted line indicates number of CTCF-disrupting variants observed in the height GWAS variants overlapping ATAC-seq regions (n = 78, p<0.01). (D) An example of a variant (rs12529733) that overlaps ATAC-seq peaks and is predicted to alter CTCF binding. Features in the figure are the same as described in *Figure 6A*.

DOI: https://doi.org/10.7554/eLife.29329.014

Our analyses revealed that ATAC-seq datasets generated from mouse growth plate chondrocytes overlap previously identified signals of chondrocyte biology in the mouse, as well as in humans. The ATAC-seq peaks reside near genes enriched for human phenotypes related to skeletal biology, such as 'metaphyseal widening.' These peaks also reside near genes with known roles in chondrocyte biology and include biological processes such as 'cartilage condensation' and 'chondrocyte proliferation.' We also show that our open chromatin profiles overlap quite faithfully with locations of active regulatory enhancers and Sox9-binding sites as assessed using ChIP-seq on mouse rib chondrocytes (*Ohba et al., 2015*). Additionally, our datasets overlap with evidence of regulatory control during chondrogenesis and human limb embryogenesis (*Cotney et al., 2013*; *Herlofsen et al., 2013*; *Kundaje et al., 2015*). These findings indicate that general aspects of chondrocyte regulation, in part shared between humans and mice, are captured by our ATAC-seq growth plate datasets. While we note that appropriate human datasets generated from growth plate chondrocytes are still unavailable and that there is evidence of substantial functional divergence in the genic and non-coding usage in cell types between species (*Bernstein and ENCODE Project Consortium, 2012*), the strong overlap between our mouse ATAC-seq data and human embryonic limb and human BMDCs suggests considerable conservation in the regulatory profiles at the growth plate.

While our datasets capture aspects of general chondrocyte biology, we also found that ATAC-seq peaks acquired on femoral tissues show enrichment in human phenotypes related to the anatomy of femur. This finding points to a fine-grained, modularized control of skeletal element-specific biology. This is also supported by our independent identification of a number of previously published long bone growth plate chondrocyte enhancers in mice and humans, as well as the growing evidence that a number of important skeletal development genes, such as *Bmp5* (*Guenther et al., 2015*), *Gdf6* (*Mortlock et al., 2003*; *Indjeian et al., 2016*), and *Gdf5* (*Capellini et al., 2017*; *Chen et al., 2016*) have conserved modularized *cis*-regulatory architectures. As height is encoded by a number of genetic inputs, undoubtedly variants that influence general, as well as element-specific regulatory enhancers (e.g. tibia versus femur), will be important factors shaping height variation in humans.

Given that over 80% of GWAS height loci are non-coding, our study demonstrates that height GWAS variants are significantly enriched in growth plate chondrocyte regulatory regions and that the majority reside in intergenic portions of the genome. Importantly, this enrichment of height GWAS loci was not observed nearly as strongly in other chondrocyte or limb bud tissues, which suggests that our dataset is capturing functional genetic variation influencing height that is not captured by other existing datasets. No enrichment was observed for DNaseI hypersensitivity sites from mouse brain, liver, or lung, which suggests that our enrichment with height GWAS variants is tissue-specific and also not due to an artifact from the liftover process from mouse to human. These ATAC-seq peaks overlapping height GWAS variants are enriched for nearby differentially expressed growth plate genes. GWAS results for other traits/diseases that are not clearly related to growth/height did not show enrichment at the ATAC-seq peaks, demonstrating that the ATAC-seq peaks are specifically capturing growth/height biology. Finally, our motif analysis additionally revealed that a number of these human height variants may alter the binding of transcription factors with important roles in chondrogenesis and gene regulatory processes.

Combining our statistical fine-mapping and functional fine-mapping has allowed us to narrow down to a much smaller subset of putatively causal variants for downstream functional testing. As an example, one putatively causal variant (rs9920291) resides in the *Chondroitin Sulfate Synthase 1* (*CHSY1*) locus. Human mutations in *CHSY1* result in Temtamy preaxial brachydactyly syndrome, characterized by short stature, preaxial brachydactyly, and hyperphalangism of digits 1–3 (45, 46), as well as a number of additional soft and hard tissue disorders. rs9920291 is in the fine-mapped credible set, resides in regulatory elements active in human adult chondrocytes and embryonic limb

tissues, and displays evidence of binding by Sox9 in mouse chondrocytes. rs9920291 additionally modifies a critical nucleotide (C to T) and disrupts the experimentally determined in vitro binding motif of HOXD13, a homeodomain transcription factor that is expressed in chondrocytes (*Kuss et al., 2014*; *Reno et al., 2016*) and has known roles in skeletal development (*Johnson and Tabin, 1997*). We performed extensive experimental characterization of rs9920291 at the *CHSY1* locus. We demonstrated that rs9920291 displays allelic regulatory activity in human chondrocyte cell lines. We also deleted the regulatory region surrounding rs9920291 using CRISPR-Cas9 and demonstrated that these deletions alter *CHSY1* expression. Thus, our results elucidate how common regulatory variation at this locus influences height by modulating *CHSY1* expression. Height, given that it is generally not a cell-intrinsic phenotype, has traditionally not been conducive to functional experimentation. Our results highlight how we can pinpoint likely causal variants and mechanisms to make experimentation more tractable for a challenging phenotype such as height. To our knowledge, along with *GDF5*, this is the only other height GWAS locus with functional validation (*Capellini et al., 2017*).

A number of other variants, listed in the browsable *Supplementary file 6*, are available for functional testing. When more extensive height GWAS data (ideally imputed to a dense reference panel) are generated and more sophisticated fine-mapping methods are developed, additional evidence of enrichment for fine-mapped variants may be revealed using our approach. When these experiments are combined with high-throughput functional assays to test variant functionality, a more refined picture of height biology will emerge.

## Materials and methods

### ATAC-seq methods

All experiments were performed following protocols approved by the Harvard University IACUC committee. Timed matings were established between FVB mice, and pregnant females were euthanized at E15.5 in order to acquire embryonic long bone growth plates. Embryos were dissected in PBS1X on ice under a dissection scope and the proximal and distal growth zones of the right and left femur were stripped clear of soft tissues. Each proximal or distal cartilaginous end was then micro-dissected from the bony diaphysis (*Figure 1*) and separately pooled from a single litter, consisting on average of eight animals. Two biological replicates were collected in line with previous ATAC-seq studies (*Gehrke et al., 2015*). The samples were collected in micro-centrifuge tubes containing 200 ul 5% FBS/DMEM. To generate a single-cell chondrocyte suspension, each pooled sample was then subjected to 1% Collagenase II (VWR 80056–222, Radnor, Pennsylavania) digestion for 2 hr at 37°C rocking, mixing every 30 min. After placing on ice, samples were filtered using a micro-centrifuge filter set-up by gently mashing the residual tissues through the filter followed by rinsing with 5% FBS/DMEM. Samples were then spun down at 500 g at 4°C for 5 min. We next performed cell counting methods using trypan blue and a hemocytometer and performed subsequent ATAC-seq steps on those samples that had cell death rates well below 25%. On average we acquired 500,000–1,000,000 living cells per harvest. Next, cells were re-suspended in concentrations of 50,000 cells in 1x PBS. Cell samples then subjected to the ATAC-seq protocol as described in *Buenrostro et al., 2015Buenrostro et al., 2015*, modifying the protocol by using 2 μl of transposase per reaction. The transposase reaction product was then purified using the Omega MicroElute DNA Clean Up Kit following manufacturers protocols, eluted in 10 μl of warmed ddH20, and stored at −20°C.

Next, samples were subjected to PCR amplification and barcoding following *Buenrostro et al., 2015Buenrostro et al., 2015*. All primers used in this step are listed in *Supplementary file 1*. Ten microliters of transposed DNA were then placed in a reaction containing NEBNext High-Fidelity PCR Master Mix, ddH20, and primers. After amplification, samples were transferred to micro-centrifuge tubes and subjected to the OMEGA Bead Purification Protocol following manufacturers protocol. The samples were eluted in 30 μl of TE, run on a nanodrop, diluted to 5 ng/μl and run on a bioanalyzer. Prior to sequencing sample concentrations were determined using the KAPA Library Quantification Complete Kit (KK4824). Samples were then sent out to the Harvard University Bauer Core Facility for sequencing on one lane of the Illumina NextSeq 500 (see *Supplementary file 1*).

Sequencing yielded approximately 400 million reads per lane and an average of 100 million per sample. Quality control statistics are presented in *Supplementary file 1*. Sequenced reads were next aligned to the mouse reference mm10 genome assembly using Bowtie2 (*Langmead and Salzberg, 2012*). Duplicated reads were removed and significant peaks determined by using MACS2 software (version 2.1.0) (*Johnson and Tabin, 1997*). Peaks were assessed for reproducibility between biological replicates using the IDR statistical test (*Li et al., 2011*) at various statistical cut-offs, with an IDR threshold of <0.05 selected to define reproducible peaks for all subsequent analyses. Datasets for other IDR score thresholds are available upon request and when used in all analyses show similar statistical enrichments. Intersections between proximal and distal femur peaks were done with IDR-filtered narrow peak files, with the proximal/distal-common set generated by merging overlapping peaks. Annotations of the narrow peak files generated on all datasets were made using HOMER annotatePeaks.pl (http://homer.salk.edu/homer/motif/).

Raw sequencing fastq files and processed peak bed files have been deposited on NCBI GEO (GSE100585).

## GREAT analyses

GREAT (Genomic Regions Enrichment of Annotations Tool) (*McLean et al., 2010*) identifies curated gene sets that display enrichment near regulatory elements. The software first identifies likely target genes of each regulatory element based on distance. GREAT then tests the enrichment of these genes localized near regulatory elements in curated gene sets that reflect pathways and molecular processes. To increase the specificity of our results, we removed peaks that were also seen in ATAC-seq from the mouse brain. ATAC-seq results were then lifted over to hg19 for analysis. We used the 'Significant By Region-based Binomial view'; otherwise, all default input parameters and output display settings available at the GREAT website (http://great.stanford.edu/) were used. GREAT v3.0.0 was used.

## Additional epigenetic datasets

Mouse chondrocyte data: ChIP-seq profiles Sox9 binding in mouse rib chondrocytes was obtained from *Ohba et al. (2015)*. We also obtained ChIP-seq of RNA Pol II, H3K4me2, H3K4me3, H3K27ac, p300 and H3K27me3 in mouse chondrocytes from *Ohba et al. (2015)*.

Human data: H3K27ac ChIP-seq of E47 human limb buds that are shared with mouse was obtained from *Cotney et al., 2013* (*Cotney et al., 2013*). H3K27ac ChIP-seq of human adult cultured BMDCs was obtained from http://egg2.wustl.edu/roadmap/web_portal/processed_data.html (*Herlofsen et al., 2013*; *Kundaje et al., 2015*). The consolidated epigenome BroadPeak file was used.

Mouse ENCODE data: DNA-seq data were downloaded from the mouse ENCODE project (*Yue et al., 2014*). All samples were from *Mus musculus* C57BL/6 and included brain male embryo (14.5 days) (ENCSR000COE), liver male adult (8 weeks) (ENCSR000CNI), and lung male adult (8 weeks) (ENCSR000CNM). Processed NarrowPeak bed files were downloaded for each tissue and then lifted over to hg19 as described above.

## Epigenetic intersection analyses

All intersection analyses of ATAC-seq peaks with GWAS loci, genes, or ChIP-seq peaks were performed using BEDTools v2.18 (*Quinlan and Hall, 2010*) and custom R scripts (R version 3.1.1) (see *Source code 1*). For overlaps of genes with GWAS loci, an overlap was called for a given height GWAS locus if the gene's transcriptional boundaries were within 100 kb of any SNP in the locus (defined as $r^2$ >0.5 to lead SNP).

For overlap of ChIP-seq datasets, any overlap (>=1 bp) was considered to represent the same feature. To evaluate the significance of overlap, a permutation-based procedure was implemented in the R package regioneR (*Gel et al., 2016*). We used the 'circularRandomizeRegions' function to generate random permutations. In this function, each chromosome is 'circularized', and the peaks bed file is shifted at random along the circularized chromosome and evaluated for overlap. We used 1000 permutations to calculate the empiric p value.

## GoShifter analysis

GoShifter v0.2 was used to evaluate the statistical enrichment of the overlap between height GWAS loci and ATAC-seq peaks (or other epigenetic peaks) (*Trynka et al., 2015*). The European (EUR) subset of 1000 Genomes Phase 3 (*Auton et al., 2015*) was used to identify proxy SNPs and calculate LD. An LD cutoff of $r^2 >0.5$ to the index SNP was used. A window size of 100 kb was used to find LD SNPs ('window' option). All other parameters were set to the software default. 100,000 permutations were run. GoShifter analyses for additional traits/diseases were performed in the same fashion.

## Generation of matched GWAS loci

SNPSNAP (*Pers et al., 2015*) was used to generate matched loci for the height GWAS loci. The European (EUR) subset of 1000 Genomes Phase 3 (*Auton et al., 2015*) was used to identify proxy SNPs and calculate LD. Loci were defined as variants with $r^2 > 0.5$ to the index SNP. Loci were matched using default criteria: MAF ± 5%, gene density ± 50%, distance to nearest gene ±50% and LD buddies ± 50%. SNPSNAP was able to generate matched gene sets for 676 out of the 697 loci from *Wood et al. (2014)*. 1000 matched sets of random GWAS loci were generated.

## Differentially expressed growth plate genes

A list of 427 differentially expressed mouse genes was downloaded from *Lui et al. (2012)*. These 427 differentially expressed genes were determined in the *Lui et al. (2012)* paper based on meeting two of three criteria: (1) spatial regulation in the growth plate, (2) temporal regulation during growth plate formation, and/or (3) specific expression in the growth plate (as compared to other tissues such as the lung, kidney, heart). Orthologous human genes and their respective positions were identified using Ensembl Biomart (*Kinsella et al., 2011*). The 'Ensembl Genes 86' database was used.

## eQTL overlap analysis

Using the set of 427 differentially expressed growth plate genes (see above), we identified 157 genes that were also genes for cis-eQTLs in whole blood (FDR < 0.05) as identified by *Westra et al. (2013)*. For each of these 157 eQTL genes, the corresponding eQTL variant with the strongest p value was used. Using this set of eQTL variants, GoShifter was run as described above to test for enrichment at ATAC-seq peaks.

## GTEx data

The Genotype-Tissue Expression (GTEx) Project (Lonsdale J, GTEx Consortium, 2013) was supported by the Common Fund of the Office of the Director of the National Institutes of Health, and by NCI, NHGRI, NHLBI, NIDA, NIMH, and NINDS. The data used for the analyses described in this manuscript were obtained from the GTEx Portal on 10/27/2017.

## Motif analysis

ATAC-seq peak centers from proximal/distal tissues were aggregated and padded outwards to a fixed length of 200 bp (based on the distribution of ATAC-seq peak sizes). HOMER (*Jerković et al., 2017*) de novo motif analysis was performed for the sequence set utilizing a 10x random shuffling as a background set. De novo motifs were compared to a vertebrate motif library provided by HOMER, which includes the JASPAR database (*Visscher et al., 2010*), with matches scored using Pearson's correlation coefficient of vectorized motif matrices, with neutral frequencies (0.25) substituted for non-overlapping positions.

Lifted over human ATAC-seq peaks were then intersected with height GWAS variants from *Wood et al. (2014)*. Variants were characterized for selected motifs which appeared in both the mm10 and hg19 de novo motifs using motifbreakR (*Coetzee et al., 2015*). Motif matches were scored using log-odds scoring (with 0.25 A/C/G/T background frequencies) for both reference and mutated sequences. Effects on position-weighted matrix (PWM) scoring were calculated as differences between the log-odds score of reference and mutant motif hits, with negative differences representing a disrupted motif. To assess enrichment for CTCF-disrupting variants in the ATAC-intersected variant set, 500 randomly sampled subsets (n = 928) of the height GWAS variants (excluding those intersecting ATAC-seq peaks) were generated and counted for CTCF-disrupting

variants. Counts were standardized and statistical significance was assessed using a continuous distribution function of the standard normal distribution.

To assess enrichment of predicted transcription factor binding near variants intersecting ATAC-seq peaks, a set of hg19 DNA sequences was first constructed using 30 bp windows centered on each variant, generating an ATAC-intersected subset and whole GWAS superset. A utility from HOMER (findmotifs.pl) was then used to scan this sequence set using its provided vertebrate motif library to search for motif hits. Motifs from JASPAR 2016 were filtered (386 motifs) and counted within each of the two sets. Hypergeometric tests were performed for all motifs using Benjamini-Hochberg FDR correction (*Benjamini and Hochberg, 1995*).

## Expression plasmids

CRISPR-Cas9 PX458 vector was obtained from Addgene (Cambridge, Massachusetts). pGL4.23 luciferase and pGL4.74 renilla vectors were acquired from Promega (Madison, Wisconsin). Plasmids encoding HOXD13-FLAG and HOXD13-HA were generously gifted from Dr. Vincenzo Zappavigna (University of Modena and Reggio Emilia) (*Caronia et al., 2003*).

## Cell lines and culture conditions

Human embryonic kidney (HEK-293FT) cells were acquired from Dr. Pardis Sabeti (Harvard University and Broad Institute). T/C-28a2 human chondrocyte cells were acquired from Dr. Li Zeng (Tufts University) courtesy of Dr. Mary Goldring (The Hospital for Special Surgery). Both cell lines were cultured at 5% $CO_2$ at 37°C in Dulbecco's Modified Eagle's Medium (DMEM), 10% fetal bovine serum (FBS), and 1% penicillin-streptomycin (P/S). Media was replaced every 2–3 days and the cells were sub-cultured every 5 days.

## CRISPR-Cas9 targeting at the *CHSY1* regulatory element

All guide RNAs surrounding the human *CHSY1* regulatory element or rs9920291 were designed using MIT CRISPR Tools (http://crispr.mit.edu), synthesized by Integrated DNA Technologies, Inc (Coralville, Iowa), and cloned into the PX458 vector following published protocols (*Ran et al., 2013*). The sequence of all sgRNAs are listed in *Supplementary file 10* and their locations with respect to the targeted *CHSY1* regulatory element and rs9920291 are found in *Figure 6—figure supplement 1*. Guide RNAs were initially tested for ability to induce efficient deletions of the human element in cultured HEK-293FT cells. After two days of culture at 37°C, transfected HEK-293FT cells were examined under a GFP-microscope to verify successful transfection and GFP expression. DNA was then extracted using E.Z.N.A Tissue DNA Kit, and the *CHSY1* regulatory element region was amplified using PCR and primers surrounding the guide RNA design sites (Primers: 'rs9920291 Forward' and 'rs9920291 Reverse', *Supplementary file 10*). Amplification products were isolated from 1% agarose gels (E.Z.N.A Gel Extraction Kit) and Sanger sequenced to verify deletions of various sizes (see below).

After confirmation that sgRNAs worked in HEK-293FT cells, we performed in vitro deletion in the T/C-28a2 chondrocyte cell line (*Kokenyesi et al., 2000*; *Finger et al., 2003*). The same sgRNAs were used as above. T/C-28a2 cells were maintained in DMEM (Gibco, Gaithersburg, Maryland) supplied with 10% FBS (Gibco) and 1% Pen/Strep (0.025%), and seeded in a six-well plate 1 day prior to transfection. After culturing at 37°C, we scanned cells under a GFP-microscope to verify successful transfection efficiency (i.e.,>70% of cells) and GFP expression (N = 3). DNA was then extracted using E.Z.N.A Tissue DNA Kit, and the *CHSY1* regulatory element region was amplified using PCR primers surrounding the guide RNA design sites (MiSeq Primers Forward and MiSeq Primers Reverse, *Supplementary file 10*). Amplification products were isolated from 1% agarose gels (E.Z.N.A Gel Extraction Kit). Mi-Seq sequencing at the MGH CCIB DNA Core was used to verify successful targeting of the larger *CHSY1* regulatory region (hg19: chr15:101,749,730-101,749,865) with a modification efficiency of approximately 10% and smaller *CHSY1* regulatory region around rs9920291 (hg19: chr15:101,749,797–101,749,813) with a modification efficiency of approximately 8.5%. RNA was also extracted from control and CRISPR-Cas9 targeted T/C-28a2 cells. RNA was DNase-treated and converted to cDNA for qPCR (see below).

## cDNA synthesis and qRT-PCR analysis

Total RNA was extracted from HEK-293FT and T/C-28a2 cells and prepared using the Trizol Reagent (Thermo Fisher Scientific, Springfield Township, New Jersey) and Direct-zolTM RNA Miniprep kit (ZYMO). Three micrograms of total RNA were used to synthesize first-strand cDNA using Super-Script III First-Strand Synthesis System (Thermo Fisher Scientific). qRT-PCR analysis was then performed with specific primers and Applied Biosystems Power SYBR master mix (Thermo Fisher Scientific) with GAPDH as a reference gene. Primers used for qRT-PCR are listed in *Supplementary file 10*.

## Cloning, construct preparation, in vitro luciferase reporter and overexpression assays

Prior to in vitro transfection of *CHSY1* regulatory sequences into HEK-293FT and T/C-28a2 cells (see below), primers containing KpnI/HindIII linker sequences were used to amplify the putative regulatory element surrounding rs9920291. (see above and *Supplementary file 10*). The PCR protocol was initiated at 98°C for 30 s, followed by 34 cycles at 98°C for 20 s, 60°C for 20 s, and 72°C for 30 s, and then followed by a 72°C for 5 min final extension. Coriell (Camden, New Jersey) sample NA12286 was used to amplify the region containing the reference allele of rs9920291 and NA19119 was used to amplify the region containing the variant allele of rs9920291. Each amplicon was then ligated into a KpnI/HindIII containing pGL4.23 firefly luciferase vector using NEBuffer 2.1 (NEB B7202S) for digestion and T4 DNA Ligase (NEB M0202S) for ligation according to manufacturer's protocols (New England Biolabs, Beverly, Massachusetts). Ligates for constructs containing inserted or non-inserted ('empty') sequence were next transformed into DH10B cells, and streaked on ampicillin plates. Single colonies were then picked, PCR screened, sequenced, and purified using E.Z.N.A. Endo-Free Plasmid DNA Maxi Kits (Omega D6926-03). The PCR protocol used for screening is listed above. Sanger sequencing was used to confirm the orientation as well as sequence identity of each insert as well as the entire pGL4.23 firefly vector. pGL4.74 renilla vector was similarly transformed, amplified, purified and sequence verified. Prior to HOXD13 over-expression experiments, HOXD13 constructs were transformed into DH10B cells, streaked on ampicillin plates, individual colonies were grown and purified via Endo-Free Plasmid DNA Maxi Kits (Omega D6926-03). Purified plasmids underwent diagnostic digest and sequence verification.

HEK-293FT cells and T/C-28a2 chondrocytes were passaged using standard conditions (see above) prior to reporter gene transfection and HOXD13 over-expression experiments. Prior to transfection, cells were first cultured in DMEM with 10% FBS for 24 hr in 96-well dishes at a seeding density of $3 \times 10^4$ cells/well. The volume of media per well was then brought to 100 µl. For luciferase reporter experiments, cells were then transiently transfected with 100 ng firefly luciferase reporter vector or empty pGL4.23 luciferase vector, along with 5ng pGL4.74 renilla vector. Transfections were performed using Lipofectamine 2000 (Invitrogen 11668–019) at a Lipofectamine:DNA ratio of 4:1 according to manufacturer instructions. Luciferase activity was measured 48 hr after transfection using the 96-welll Dual-Glo Luciferase Assay System (Promega E2940) following manufacturer protocols on a SpectraMax L Microplate Reader (Molecular Devices; Cat# SpectraMax L Config) with a 1 min dark adapt, 5 s integration, and max range settings. For cell culture experiments, to compare expression between the reference and alternate allele *CHSY1* constructs (*Figure 6b*), we performed at least four independent transfection experiments at each concentration containing eight technical replicates (i.e. individual wells of a 96-well plate) per construct. We first normalized each firefly luciferase value per well by its corresponding renilla luciferase value per well, and then compared the mean expression of the reference allele construct (normalized by empty vector) to that of the mean expression of the alternate allele construct (normalized by empty vector) using a two-sample directional Student's t-test. For HOXD13 overexpression experiments, HEK-293FT and T/C-28a2 cells were transfected as above but with 0, 2 and 4 µg of HOXD13 expression or control constructs (N = 3 biological replicates per condition), and after 24, 48, and 72 hr RNA was collected for qRT-PCR experiments (see above).

## CHSY1 allelic skew analyses

Both HEK-293FT cells and T/C-28a2 chondrocytes were screened using Sanger sequencing to identify genotype rs9920291. Sanger sequencing results confirmed preliminary findings using the HEK-

293 browser (http://hek293genome.org/v2/) that HEK-293FT cells were heterozygous at rs9920291, unlike T/C-28a2 cells which were homozygous for the reference C allele. Thus, only HEK-293FT cells could be used for allele-specific expression. Next, using the HEK-293 browser with follow-up verification using Sanger sequencing, we identified two heterozygous transcript variants (exon 3 coding variant [rs28364839] and 3'UTR variant [rs11433]).

For allelic skew analyses on untreated HEK-293FT samples, five independent biological replicates were grown and harvested. RNA was isolated separately from each sample using the Trizol Reagent (15596–026, Ambion by Life Technologies, Norwalk, Connecticut) and the RNA Clean and Concentrator™-5 Kit (supplied with DNase I, Zymo). Samples were then run on a bioanalyzer to achieve RNA Integrity Numbers greater than 8. These RNA samples were then reverse transcribed using a SuperScript IV First Strand cDNA Synthesis Reaction kit (18090010, Life Technologies) following manufacturer's recommendations.

cDNA samples were then sent to EpigenDx for allelic skew analysis assay design and execution. SNPs in the coding regions of *CHSY1* (rs11433 and rs28364839), were validated by EpigenDx (Hopkinton, Massachusetts). Pyrosequencing for SNP genotyping (PSQ H96A, Qiagen Pyrosequencing) is a real-time sequencing-based DNA analysis that quantitatively determines the genotypes of single or multiple mutations in a single reaction. Briefly, 1 ng of sample cDNA was used for PCR amplification. PCR was performed using 10X PCR buffer (Qiagen Inc., Maryland) at 3.0 mM MgCl2, 200M of each dNTP, 0.2 µM each of the forward and reverse primers (available through EpigenDx), and 0.75 U of HotStar DNA polymerase (Qiagen Inc.), per 30 µl reaction. PCR cycling conditions were: 94°C 15 min; 45 cycles of 94°C 30 s; 60°C 30 s; 72°C 30 s; 72°C 5 min. One of the PCR primer pairs was biotinylated to convert the PCR product to single-stranded DNA sequencing templates using streptavidin beads and the PyroMark Q96 Vacuum Workstation. Ten microliters of the PCR products were bound to streptavidin beads and the single strand containing the biotinylated primer was isolated and combined with a specific sequencing primer (available through EpigenDx). The primed single stranded DNA was sequenced using the Pyrosequencing PSQ96 HS System (Qiagen Pyrosequencing) following the manufacturer's instructions (Qiagen Pyrosequencing). The genotypes of each sample were analyzed using Q96 software AQ module (Qiagen Pyrosequencing). Pyrosequencing results for each *CHSY1* exonic or 3'UTR SNP were used to calculate the allelic ratios in the heterozygous state.

## Statistical analysis

For regulatory element transfection experiments, CRISPR-Cas9 deletion experiments, and HOXD13 over-expression experiments, the means ± SEM of multiple independent measurements were calculated. The unpaired two-tailed Student's t test was used to determine the significance of differences between means. P values smaller than 0.05 (*p<0.05, **p<0.01, ***p<0.001) were considered to be statistical significant.

## UniProbe analysis

To find upstream transcription factors (TF) predicted to bind at rs9920291 near *CHYS1* we used UniProbe (*Hume et al., 2015*). UniProbe (http://the_brain.bwh.harvard.edu/uniprobe/about.php) is built on experimental measurements of binding affinities between large numbers of expressed TFs and all possible 8-mer target oligonucleotides (*Hume et al., 2015*). For two 15 base-pair sequences, each centered at rs9920291 but different only at the C/T variant (Reference [REF] TGCTAAACAAAAATA and Alternate [ALT] TGCTAAATAAAAATA), we imported each sequence separately into the 'Search for TF Binding Sites' window on the UniProbe browser (http://the_brain.bwh.harvard.edu/uniprobe/) with the following conditions: Score Threshold: 0.4; Species: Homo sapiens. We acquired 18 TF predictions for REF and 61 TF predictions for ALT. We next imported the results into *Supplementary file 7* (sheet 1: REF TGCTAAACAAAAATA and sheet 2: ALT TGCTAAATAAAAATA) and performed an intersection (sheet 3: REF (red) vs. ALT (blue)) to identify those TF-binding preferences that were reduced/gained between the two sequences. For a TF-binding site to be considered reduced or gained, the site must exhibit an enrichment score change of greater than at least 0.10, which corresponded to significant changes in z-score (e.g., *Figure 6a*). To identify the change in HOXD13 binding affinity, we re-ran UniProbe on the REF sequence but at a lower enrichment score of E >= 0.2. This yielded 341 TF predictions (sheet 4), which were then compared to the 61 TF

predictions generated at E >= 0.4 (sheet 5). Finally, to generate the z-score, we downloaded HOXD13 experiment information from UniProbe (http://the_brain.bwh.harvard.edu/uniprobe/browse.php) and identified the corresponding z-score for each 8-mer.

Lists of potential upstream regulators exhibiting marked reduction/gain in binding affinity were then screened using expression and phenotypic data to identify those TFs also expressed or functionally required in growth plates. To carry out this analysis, we used data in VisiGene (http://genome.ucsc.edu/cgi-bin/hgVisigene), Eurexpress (http://www.eurexpress.org/ee/), Genepaint (http://www.genepaint.org/Frameset.html), and the Mouse Genome Informatics expression and phenotypic databases (http://www.informatics.jax.org). HOXD13 met the above criteria. HOXD13 had enrichment scores below the recommended UniProbe threshold of 0.4 for the REF sequence but much higher (above 0.4) for the ALT sequence, and this corresponded to a significant change in z-score (*Figure 6a*).

## Acknowledgements

The authors would like to thank the following individuals for assistance with this manuscript: Dr. William Greenleaf and Dr. Jason Buenrostro for advice on ATAC-seq.; Dr. Claire Reardon and the entire Harvard University Bauer Core facility for assistance with next generation sequencing; Dr. Michelle Clamp at Harvard University Research Computing for assistance with ATAC-seq data; Dr. Shinsuke Ohba for providing access to Sox9 ChIP-seq datasets; Dr. Lin Wu for assistance with mouse stable line generation; Dr. Vincenzo Zappavigna for HOXD13 expression constructs; Drs. Li Zeng and Mary Goldring for T/C-28a2 chondrocytes; Dr. Pardis Sabeti for HEK-293FT cells. This work was funded in part by the Harvard University Milton Fund and NSF (BCS-1518596) to TDC, and National Institutes of Health (R01 DK075787) to JNH.

## Additional information

### Funding

| Funder | Grant reference number | Author |
| --- | --- | --- |
| National Science Foundation | BCS-1518596 | Terence D Capellini |
| National Institutes of Health | R01 DK075787 | Joel Hirschhorn |

The funders had no role in study design, data collection and interpretation, or the decision to submit the work for publication.

### Author contributions

Michael Guo, Conceptualization, Data curation, Formal analysis, Investigation, Visualization, Methodology, Writing—original draft, Writing—review and editing; Zun Liu, Cameron P Shaw, Formal analysis, Investigation; Jessie Willen, Evelyn Jagoda, Data curation, Investigation, Methodology; Daniel Richard, Formal analysis, Investigation, Writing—original draft, Writing—review and editing; Andrew C Doxey, Methodology, Writing—original draft; Joel Hirschhorn, Supervision, Funding acquisition, Writing—review and editing; Terence D Capellini, Conceptualization, Supervision, Funding acquisition, Investigation, Writing—original draft, Project administration, Writing—review and editing

### Author ORCIDs

Michael Guo (iD) https://orcid.org/0000-0002-1357-6389
Terence D Capellini (iD) https://orcid.org/0000-0003-3842-8478

### Ethics

Animal experimentation: This study was performed in strict accordance with the recommendations in the Guide for the Care and Use of Laboratory Animals of the National Institutes of Heath, and under protocols approved by the Institutional Animal Care and Use Committee (IACUC) at Harvard University (Protocol # 13-04-161). Euthanasia was conducted according to the recommendations and

guidelines from the Panel on Euthanasia of the American Veterinary Medical Association and as outlined in Protocol #13-04-161.

## Decision letter and Author response

Decision letter https://doi.org/10.7554/eLife.29329.036
Author response https://doi.org/10.7554/eLife.29329.037

## Additional files

### Supplementary files

• Source code 1.
DOI: https://doi.org/10.7554/eLife.29329.015

• Supplementary file 1. Description of each ATAC-seq dataset and its QC data. Column headings are self-explanatory. See Materials and methods for details.
DOI: https://doi.org/10.7554/eLife.29329.016

• Supplementary file 2. GREAT output using proximal femur ATAC-seq dataset. Data used to generate this table are hg19 liftover ATAC-seq coordinates from the entire proximal femur ATAC-seq analysis. Table shows top 50 hits per each category. See great.stanford.edu for details on column headings.
DOI: https://doi.org/10.7554/eLife.29329.017

• Supplementary file 3. GREAT output using distal femur ATAC-seq dataset. Data used to generate this table are hg19 liftover ATAC-seq coordinates from the entire distal femur ATAC-seq analysis. Table shows top 50 hits per each category. See great.stanford.edu for details on column headings.
DOI: https://doi.org/10.7554/eLife.29329.018

• Supplementary file 4. GREAT output using union femur ATAC-seq dataset. Data used to generate this table are hg19 liftover ATAC-seq coordinates from the entire union (proximal plus distal) femur ATAC-seq analysis. Table shows top 50 hits per each category. See great.stanford.edu for details on column headings.
DOI: https://doi.org/10.7554/eLife.29329.019

• Supplementary file 5. Collection of published chondrocyte enhancers and their intersection with ATAC-seq data. Columns 1–5: Data on the location (chrom, start and end), gene and reference for published growth plate chondrocyte enhancers in mouse and humans. Column 6: Data on the location of overlapping proximal and distal femur union ATAC-seq set in mm9 coordinates.
DOI: https://doi.org/10.7554/eLife.29329.020

• Supplementary file 6. Height variants overlapping ATAC-seq peaks. Columns: Lead rsID and Lead SNP (chr:position:reference:alternate) represent information for each index SNP from *Wood et al., 2014*. Similarly, LD rsID and LD SNP represent information for each proxy SNP ($r^2$ >0.5 to lead SNP). $r^2$ shows the LD correlation between the lead SNP and proxy SNP. GWAS p value shows the p value from *Wood et al., 2014* for the lead SNP. PICS Score represents the credible set probability. PICS Credible Set shows whether that SNP is present in the 95% credible set (1 for present, 0 for absent). ATAC-seq distal, proximal, and union show whether the proxy SNP overlaps a respective ATAC-seq peak lifted over to human. For proxy SNPs that do overlap, the position of the ATAC-seq peaks is shown; otherwise, 'NA' represents lack of overlap. Expressed genes represents whether there is a differentially expressed growth plate gene (±100 kb) from the locus based on data from *Lui et al., 2012*. All positions are based on the human hg19 reference genome.
DOI: https://doi.org/10.7554/eLife.29329.021

• Supplementary file 7. Motif analysis of rs9920291 in the *CHSY1* locus using UniProbe. See Materials and methods for details.
DOI: https://doi.org/10.7554/eLife.29329.022

• Supplementary file 8. HOMER de-novo motif analysis results for mm10 and hg19 ATAC-seq peaks Columns: Motif Logo: Logo of generated de novo motif. Rank: Motifs ranked by calculated p value, p value and log p value: p values for motif enrichment calculated by HOMER. % of Targets/Background: Percent of target and background sequences predicted to contain this motif. STD(Bg STD):

Standard deviation of motif occurrence (bp) away from the center of the 200 bp sequence for target and background sets, respectively. Similar Motif: Similar motif as identified through search of HOMER vertebrate motif library; similarity score shown in parentheses (see Materials and methods).
DOI: https://doi.org/10.7554/eLife.29329.023

• Supplementary file 9: Transcription factor motif disruption by height GWAS SNPs Height GWAS variants within ATAC-seq peaks (lifted to human hg19) which intersected motifs were counted as the foreground set. All height GWAS variants (regardless of whether they intersect ATAC-seq peaks) which intersected motifs were counted as the background set (a super-set of the foreground windows). For a given transcription factor (e.g. PBX1) all hits for the corresponding motif within the foreground and background windows were counted ('Foreground Motif Hits' and 'Background Motif Hits', respectively), along with the total number of all motif hits in the foreground set ('All Foreground Motif Hits') and the number of additional motif hits in the background set ('Remainder Background Motif Hits', calculated as the total number of all motif hits in the background windows minus the number of motif hits for the given transcription factor in the background windows). Hypergeometric test p values ('pval') and Benjamini-Hochberg FDR corrected p values ('adjusted_pval') are reported.
DOI: https://doi.org/10.7554/eLife.29329.024

• Supplementary file 10. Table of all primers and sgRNAs.
DOI: https://doi.org/10.7554/eLife.29329.025

• Transparent reporting form
DOI: https://doi.org/10.7554/eLife.29329.026

### Major datasets
The following datasets were generated:

| Author(s) | Year | Dataset title | Dataset URL | Database, license, and accessibility information |
|---|---|---|---|---|
| Guo MH, Willen J, Richard D, Jagoda E, Doxey A, Hirschhorn JN, Capellini TD | 2020 | Epigentic profiling of murine proximal and distal femoral growth plates at E15.5 of mouse gestation | https://www.ncbi.nlm.nih.gov/geo/query/acc.cgi?acc=GSE100585 | Publicly available at the NCBI Gene Expression Omnibus (accession no: GSE100585) |
| Cotney J, Leng J, Yin J, Reilly SK, et al | 2013 | The Evolution of Lineage-Specific Regulatory Activities in the Human Embryonic Limb | https://www.ncbi.nlm.nih.gov/geo/query/acc.cgi?acc=GSE42413 | Publicly available at the NCBI Gene Expression Omnibus (accession no: GSE42413) |
| Bernstein BE, Stamatoyannopoulos JA, Costello JF, Ren B, et al | 2013 | BI Human Reference Epigenome Mapping Project | https://www.ncbi.nlm.nih.gov/geo/query/acc.cgi?acc=GSE17312 | Publicly available at the NCBI Gene Expression Omnibus (accession no: GSE17312) |
| Ohba S, He X, Hojo H, McMahon AP | 2015 | Distinct Transcriptional Programs Underlie Sox9 Regulation of the Mammalian Chondrocyte. | https://www.ncbi.nlm.nih.gov/geo/query/acc.cgi?acc=GSE69111 | Publicly available at the NCBI Gene Expression Omnibus (accession no: GSE69111) |

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
