## [Decision Letter]

Thank you for submitting your article "Epigenetic profiling of growth plate chondrocytes sheds insight into regulatory genetic variation influencing height" for consideration by *eLife*. Your article has been favorably evaluated by Mark McCarthy (Senior Editor) and three reviewers, one of whom is a member of our Board of Reviewing Editors.

The reviewers have discussed the reviews with one another and the Reviewing Editor has drafted this decision to help you prepare a revised submission.

Summary:

All the reviewers were enthusiastic about your comprehensive generation of epigenetic data for the growth plate and the potential utilization for future studies on genetic regulatory regions. The ATAC-seq approach and the overlay of GWAS data will provide significant insights into murine and possibly human growth determinants.

There were three major concerns that need to be addressed before a final decision can be made.

First delineation of sex from the E15.5 murine chondrocytes and justification for the E15.5 time point are essential for understanding the framework for growth variants and for future studies by other investigators.

Second, the manuscript still lacks a strong connection to chondrocyte biology as none of the links established by the data directly address (or explain) chondrocyte proliferation or hypertrophy and genes well known to regulate them positively or negatively (PTHrp, Indian hedgehog, FGFs, IGFs, etc.).

Third, although different genes have been identified as possibly being putative regulatory variants there are no experimental or computational evidence implicating the ATAC-seq peaks in regulating the nearby growth plate genes, nor altering transcriptional regulatory activity.

These issues underlie a perception that, independent of your Nature Genetics paper, there is a lack in this manuscript of mechanistic evidence that the epigenomic profiles can be linked to functional correlates. For example, there is a suggestion that rs9920291 disrupted HOXD13 binding, but there is no evidence that HOXD13 actually binds this site at the endogenous locus. As computationally predicted TF motifs are prevalent in the genome, it is hard to assess whether this SNP/motif overlap is biologically meaningful. This requirement for a mechanistic/functional connection may require additional work.

These major points will need to be addressed in the revision along with several other points imbedded in the three reviewers’ comments that have been included below in the comments, and should be considered carefully.

Reviewer #1:

In this paper, Capellini and colleagues utilize a novel method (ATAC-seq) to read the open chromatin regulatory landscape of growth plate chondrocytes in neonatal mice. They were then able to analyze height GWAS data in the context of femur growth to gain new insights into how height GWAS variants act at the growth plate. Briefly they showed that ATAC-seq datasets generated from mouse growth plate chondrocytes overlap with previously identified signals of chondrocyte biology in the mouse, as well as in humans. They identified locations of active regulatory enhancers and Sox9 binding sites as assessed using ChIP seq on mouse rib chondrocytes and provided datasets that overlap with evidence of regulatory control during chondrogenesis and human limb embryogenesis. Their work is a nice supplement to their Nature Genetics paper identifying an ancestral regulatory variant in 'Grow'1 enhancer in mouse limbs that also affects human chondrocyte biology. Finally, the authors detected a variant in the chondroitin sulfate synthase 1 gene which might influence binding of HOXD13, a trans acting regulator of chondrocyte growth.

Strengths: There are several strengths to this paper: First the authors were able to provide very comprehensive data sets on the epigenetic landscape of mouse, and human chondrocytes, an important breakthrough in an otherwise very difficult organ to characterize. Second, they have provided evidence that these regions are critically important for growth of the chondrocyte and thus provide support from previous studies of candidate genes in chondrogenesis such as GDF5, 6, Bmp5 etc. that others have defined. Third they have been able to overlay height GWAS variants that are also significantly enriched in growth plate chondrocyte regulatory regions, particularly for those that are intergenic. Finally, this is state of the art technology in epigenetics; i.e. the use of ATAC methodology, importantly it requires fewer chondrocytes to peruse the chromatin landscape thus breaking new ground.

Limitations: – there are some concerns that need to be fully addressed:

1) The authors used E15.5 embryos from FVB mice to study the epigenetic regions for growth plate regulation. But inbred strains often exhibit numerous nucleotide variants; since C57BL6J mice are the most frequently utilized strain for in vitro and in vivo chondrocyte (and osteoblast) studies, have the authors determined whether the specific nucleotide sequence in the CHYS1 variant, rs9920291, is also present in B6 mouse chondrocytes?

2) Sex is a major determinant of height particularly early in life and during puberty. The early influences of sex steroids likely begins during late embryonic development; i.e. after E11.5 when SRY is expressed in the males, but not females. It would be important to fully understand the epigenetic landscape in respect to sex differences in vivo. Hence, the embryos should have been typed for sex, and both male and female embryos should be examined to more fully understand the transcriptional regulation of chondrocyte differentiation.

3) Can the authors justify why they did not follow up with functional studies on the CHYS1 variant rs9920291 which they identified as being a cis acting site for HOXD13.

4) Figure 6 of the E15.5 embryo showing the lacZ reporter from the ancestral GROW1 enhancer appears nearly identical to their Nature Genetics paper Figure 4; Can the authors defend its use in this manuscript? Otherwise it should be deleted.

*Reviewer #2:*

In this study, the authors have performed epigenetic profiling of active chromatin regions in E15.5 mouse embryo femurs to establish possible links with GWAS variants associated with human height previously described by other groups. The regions were found to positively associate with known genes regulating chondrocyte and skeletal biology, particularly those encoding growth plate transcription factors. These and other data lead the authors to conclude that integration of epigenetic information with genetic association data can identify mechanisms important for human height.

The study comes from a research group with recognized expertise in genetics and genomics and for important contributions to the field. It also addresses human height that can be severely affected by congenital conditions, thus introducing an element of translational medicine relevance to the work. The extensive data combining epigenetic profiling using ATAT-seq with several bioinformatics and analytical tools provide a wealth of possible links with specific genes and transcriptome data sets from previous studies. Despite these positive assets, the study raises several concerns.

While the study is technically strong, it is deficient in skeletal cell and developmental biology, raising questions about the major conclusions reached and specific relevance to human height.

The growth plate is certainly the key structure determining long bone and vertebral elongation and thus, body height. It is clearly established that the rates of skeletal growth are determined by chondrocyte proliferation (minor contribution) and chondrocyte hypertrophy (major contribution) in the growth plate. It is also well established that different growth plates within the same organism at any given time "grow" at very different rates, such as proximal and distal growth plates in radius and ulna. None of the links established by the data presented here directly address (or explain) chondrocyte proliferation or hypertrophy and genes known to regulate them positively or negatively (PTHrp, Indian hedgehog, FGFs, IGFs, etc.). The only links established are to general aspects of chondrocyte and skeletal biology, thus making it very difficult to understand how the links have anything to do with "height". The behavior of the growth plate is controlled by intrinsic mechanisms (see above) as well as systemic factors such as growth hormone. Again, none of the links related to this or other systemic cues.

The authors made use of the proximal and distal portions of E15.5 mouse embryo femur and associated perichondrium as starting material for the epigenetic analysis. The tissues were digested into single cell suspension enzymatically before processing for ATAT-seq. There is no explanation of why this developmental stage was chosen and why both ends of the femur were considered. Also, the lengthy enzymatic digestion could have altered gene expression and chromatin configuration.

To establish stronger links, the authors considered the most compelling GWAS variants overlapping ATAT-seq peaks, finding 59 variants distributed over 26 loci. The two examples mentioned are CHSY1 and HOXD13 neither of which has established roles in growth plate function. Indeed, most evidence indicates that HOXD13 regulates autopod skeletal patterning and morphogenesis, but not rates of growth.

The authors make a strong case that the data can provide insights into the meaning and functional relevance of GWAS correlations, but I do not see how they provide any direct insights into what regulates human height. It may be worth considering an overall change in rationale for the study.

*Reviewer #3:*

The article by Guo et al. describes the epigenetic profiling of mouse growth plate chondrocytes to elucidate potential molecular mechanisms through which genetic variants contribute to human height. The authors perform ATAC-seq profiling to identify open chromatin regions from murine femoral growth plates and demonstrate that femur open chromatin are (i) enriched nearby skeletal development genes, and (ii) enriched for human height GWAS loci, above other epigenomic datasets available from mouse and human. While it has previously been shown that height GWAS loci are enriched nearby musculoskeletal genes (e.g. Figure 3 of Wood et. al 2014), the identification of musculoskeletal regulatory elements possibly affected by height loci is valuable for future studies. The authors proceed to identify height GWAS loci and specific SNPs that overlap femur open chromatin peaks, and provide potential hypotheses for how select SNPs (e.g. rs9920291) may act to influence human height. Unfortunately, despite claiming the identification of "compelling mechanisms for GWAS variants" (Abstract), the authors do not provide sufficient experimental or computational evidence in support, except for rs4911178 that references data in Capellini et al. 2017 (PMID 28671685). In summary, the authors present an important epigenomic dataset that can act as a first step for future study of height GWAS SNPs, but do not provide "strong evidence" (subsection “Known and novel targets of human height variants in putative chondrocyte regulatory regions”, first paragraph) for compelling mechanisms at any height GWAS loci beyond referencing their previously published paper.

1) Substantially fewer GWAS loci overlap ATAC-seq peaks when using the fine-mapped 95% credible variants, instead of all SNPs with r^2^>0.5 (e.g. 192 loci vs. 317 loci, subsection “Fine-mapped height variants overlap with femoral growth plate open chromatin regions” and subsection “Human height variants are enriched in femoral open chromatin regions”, last paragraph; also 26 loci vs. 46 loci, subsection “Known and novel targets of human height variants in putative chondrocyte regulatory regions”, first paragraph and subsection “A subset of human height variants in femoral open chromatin regions reside near genes differentially expressed in the growth plate”, second paragraph). Given that r^2^>0.5 is not a strict LD threshold (e.g. Wood et al. 2014 uses r^2^>0.8 for strict LD), is it still meaningful to report overlap statistics using the r^2^>0.5 set when the PICS 95% credible set is available?

2) The authors state "44 different genes have been identified as possibly being modulated by these putative regulatory variants". However, in the present manuscript the authors do not provide experimental or computational evidence implicating the ATAC-seq peaks in regulating the nearby growth plate genes, and provide limited evidence that any of these overlapping variants can alter transcriptional regulatory activity.

3) Following point #2, and to support the authors' claim of identifying "prime targets for functional testing", it would be helpful to know how many GWAS SNP overlaps with femur open chromatin regions the authors believe to be biologically meaningful versus coincidence.

4) Similar to point #2, it is premature to state "59 variants thus have strong evidence for being potentially causal variants". For the authors' example of the CHSY1 locus, while there are 4 SNPs in the PICS 95% set that overlap an ATAC-seq peak nearby CHSY1, Supplementary file 6 reveals there are 87 SNPs in the 95% credible set at the CHSY1 locus. Without more evidence, it is difficult to agree with the potential causal variant conclusion.

5) Subsection “Known and novel targets of human height variants in putative chondrocyte regulatory regions”, first paragraph: The authors suggest rs9920291 disrupts HOXD13 binding, but do not provide evidence that HOXD13 actually binds this site at the endogenous locus. As computationally predicted TF motifs are prevalent in the genome, it is hard to assess whether this SNP/motif overlap is biologically meaningful.

6) Wood et al. (2014) found that height GWAS loci were more likely to be eQTLs (in blood) for genes expressed in cartilage. Did the authors examine whether the 468 credible SNPs overlapping femur open chromatin peaks are enriched for eQTL signals? This analysis may strengthen the authors' mechanistic claims.

---

## [Author Response]

[…] There were three major concerns that need to be addressed before a final decision can be made.First delineation of sex from the E15.5 murine chondrocytes and justification for the E15.5 time point are essential for understanding the framework for growth variants and for future studies by other investigators.

We thank the Reviewing Editor for summarizing these two main concerns and address each below separately.

While we agree that sex is a major determinant of height in humans particularly during puberty, our method of cell collection for ATAC-seq is not yet conducive to examining this aspect of biology. We have optimized our protocol to acquire large numbers of viable chondrocytes that have not been removed from their in vivocontext for extended periods of time.

First, we perform our ATAC-seq on freshly acquired chondrocytes, extracted from the embryonic growth plates, and we must pool embryonic long bone ends across a litter to collect enough for each specific growth plate end (i.e. proximal versus distal femur). In the process, any significant delay in extraction can significantly impact chromatin state, as the cells are no longer fresh and instead are subjected to prolonged exposure to artificial media and in vitro conditions. Genotyping for sex constitutes a major delay in the process.

Second, we perform our ATAC-seq on E15.5 embryos, a stage when sex differences are not readily apparent in the lengths of the skeletal elements (note: embryonic mice in general are not proper for studying sex-specific effects in the skeleton). However, we intentionally chose E15.5 for a number of reasons (see next point below), most notably our ability to easily extract large numbers of viable chondrocytes from the limited extracellular matrix environment using a brief collagenase digestion step (<2 hours). Our current attempts to extract chondrocytes at much later stages (i.e., E18.5 or post-natal stages – when sex differences begin to become apparent in mice) require at least 12 hours of collagenase digestion, but result in an increased ECM content that interferes with ATAC-seq and more importantly a significant increase in cell death (>25%), which results in low quality and biased ATAC-seq libraries. In addition, the prolonged duration of exposure to collagenase and artificial media in vitro that comes hand-in-hand with studying these later stages likely impacts chromatin profiles. In time, when chondrocyte extraction techniques improve, we plan to extract chondrocytes from single early post-natal mice, genotype each embryo for sex while we perform ATAC-seq, and explore sex-specific differences in growth plate chondrocyte chromatin biology.

We chose E15.5 as a stage to explore chondrocyte growth plate biology for a number of reasons:

First, growth plates at this stage show a very limited amount of extracellular matrix, which can interfere with the ATAC-seq protocol. For example, extensive extracellular matrix, characteristic of later fetal and post-natal stages, requires long periods of collagenase digestion (e.g., greater than 12 hours) to isolate chondrocytes, which in turn has a significant impact on the nature of open chromatin profiles in the resultant extracted cell populations. Increased collagenase digestion also increases cell death which also influences chromatin state biology.

Second, growth plates are readily dissected at E15.5 since soft tissues, such as tendon, ligament, and muscle, strip easily off the growth plate using simple mechanical force rather than enzymatic digestions that can alter chromatin state profiles. At later stages, increased enzymatic concentrations and processing times, as well as more mechanical force are necessary to strip soft tissues off of growth plates, and the effects of these treatments on ATAC-seq libraries remain

problematic and unclear.

Third, growth plates at E15.5 display all major chondrocyte differentiated cell types (i.e., spanning resting, proliferative, pre-hypertrophic, hypertrophic, along with perichondrial zones) and at the same time lack evidence in and around the epiphysis of secondary ossification center formation. Thus, we when we extract at this stage, we are less concerned about introducing additional cell types (e.g., osteoblasts).

Fourth, we chose to separate both long bone ends because the growth plates of the proximal and distal femur mature at different stages and under different processes, with the former undergoing complex chondrogenesis and osteogenesis patterns not evident in the latter (Stern et al., PLoS Biol. 2015; Cole et al., Bone.2013), and because proximal and distal growth plates show evidence of differential growth across a number of species, including humans. Thus, by separating growth plates, we are able to identify ATAC-seq peaks unique to each region and likely important for the marked distal femur elongation seen in humans.

Second, the manuscript still lacks a strong connection to chondrocyte biology as none of the links established by the data directly address (or explain) chondrocyte proliferation or hypertrophy and genes well known to regulate them positively or negatively (PTHrp, Indian hedgehog, FGFs, IGFs, etc.).

Thank you for raising this point regarding connections to known chondrocyte biology. In our revised manuscript, we have included ATAC-seq peaks at three well known growth plate genes: *Col2a1* (encoding a collagen subunit)*, Pth1r* (parathyroid hormone receptor), and *Fgfr1* (fibroblast growth factor)(new Figure 1—figure supplement 1). Our GREAT analysis related to human phenotypes demonstrates many known connections to growth plate-related phenotypes (new Figure 1). We now include GO Biological Processes from our GREAT analyses, which highlight terms such as “Cartilage condensation”, “Positive regulation of chondrocyte differentiation”, and “Chondrocyte proliferation”. Together, we believe these results highlight how the ATAC-seq peaks are capturing genes with fundamental roles in chondrocyte biology.

Third, although different genes have been identified as possibly being putative regulatory variants there are no experimental or computational evidence implicating the ATAC-seq peaks in regulating the nearby growth plate genes, nor altering transcriptional regulatory activity.

We thank the reviewers for their suggestions of including experimental validation. To this end, we have included a variety of experiments aimed at validating our experimental predictions at the *CHSY1* locus.

1) First, using a luciferase reporter assay, we demonstrated that rs9920291 demonstrates allelic regulatory activity in the T/C-28a2 human chondrocyte cell line (new Figure 6).

2) Using a human cell line (HEK-293FT), we demonstrate that there is allelic skew in *CHSY1* expression. We chose to use HEK-293FT because it is heterozygous “C/T” for rs9920291. We had genotyped several human chondrocyte cell lines, and unfortunately they were all homozygous “C” for this variant (new Figure 6). This is coupled with our analysis of GTEx data showing that rs9920291 is as an eQTL for *CHSY1* in human transformed fibroblasts (new Figure 6—figure supplement 1).

3) Next, we used two separate pairs of CRISPR guide RNAs to delete the region surrounding rs9920291 in the T/C-28a2 cell line. We deleted either a large 135 bp region, or a smaller 17 bp region. Deletion of either region enhances *CHSY1* expression, suggesting that rs9920291 marks a regulatory element with repressive activity (new Figure 6).

4) Finally, we demonstrate that overexpression of HOXD13 can increase *CHSY1* expression, providing evidence to support our computational predictions that HOXD13 may bind at the open chromatin region marked by rs9920291 and regulate *CHSY1* expression (new Figure 6 and new Figure 6—figure supplement 1).

Together, we believe these new results provide strong evidence that our approach can pinpoint plausible candidate causal GWAS variants. Additionally, to our knowledge, this is one of the few height GWAS loci (along with *GDF5* from our group) with experimental validation, again highlighting the novelty of our paper.

These issues underlie a perception that, independent of your Nature Genetics paper, there is a lack in this manuscript of mechanistic evidence that the epigenomic profiles can be linked to functional correlates. For example, there is a suggestion that rs9920291 disrupted HOXD13 binding, but there is no evidence that HOXD13 actually binds this site at the endogenous locus. As computationally predicted TF motifs are prevalent in the genome, it is hard to assess whether this SNP/motif overlap is biologically meaningful. This requirement for a mechanistic/functional connection may require additional work.These major points will need to be addressed in the revision along with several other points imbedded in the three reviewers’ comments that have been included below in the comments, and should be considered carefully.Reviewer #1:[…] 1) The authors used E15.5 embryos from FVB mice to study the epigenetic regions for growth plate regulation. But inbred strains often exhibit numerous nucleotide variants; since C57BL6J mice are the most frequently utilized strain for in vitro and in vivo chondrocyte (and osteoblast) studies, have the authors determined whether the specific nucleotide sequence in the CHYS1 variant, rs9920291, is also present in B6 mouse chondrocytes?

Thank you for raising this important point regarding mouse strains and the potential for there to be genetic and epigenetic differences across strains. We chose the FVB strain for this study given the consistently large number of embryos that the FVB strain produces, which allowed us to increase the number of mice from which we could collect difficult to extract tissue. We also wish to clarify that rs9920291 is a human variant that, to our knowledge, is not present in mouse. We note that the reference “C” allele is the ancestral allele that is present on the mouse sequence.

2) Sex is a major determinant of height particularly early in life and during puberty. The early influences of sex steroids likely begins during late embryonic development; i.e. after E11.5 when SRY is expressed in the males, but not females. It would be important to fully understand the epigenetic landscape in respect to sex differences in vivo. Hence, the embryos should have been typed for sex, and both male and female embryos should be examined to more fully understand the transcriptional regulation of chondrocyte differentiation.

Thank you for raising this important point regarding sex of the mice used for analyses. We have addressed this point in our response to the Editor’s first comment (see above).

3) Can the authors justify why they did not follow up with functional studies on the CHYS1 variant rs9920291 which they identified as being a cis acting site for HOXD13.

As suggested by multiple reviewers, we have pursued functional studies of our candidate causal variant rs9920291 at *CHSY1*. We describe the new experiments we have performed in our response to the Editor’s third comment (see above).

4) Figure 6 of the E15.5 embryo showing the lacZ reporter from the ancestral GROW1 enhancer appears nearly identical to their Nature Genetics paper Figure 4; Can the authors defend its use in this manuscript? Otherwise it should be deleted.

We thank the reviewer for this comment. We wish to clarify that the lacZ reporter used in this paper is from a stable line, and thus different from the original Nature Genetics paper. Given that the results are very similar, we have decided to remove the *GDF5* experiment from our paper. Additionally, we have moved the epigenetic view of the *GDF5* locus to the new Figure 6—figure supplement 2.

Reviewer #2:[…] While the study is technically strong, it is deficient in skeletal cell and developmental biology, raising questions about the major conclusions reached and specific relevance to human height.

Thank you for your careful review of our manuscript. We are pleased that you found our paper “technically strong” and with “positive assets”. Below, we address the criticisms that you raise. We hope that our revised manuscript and new experiments address these gaps and make our manuscript suitable for publication.

The growth plate is certainly the key structure determining long bone and vertebral elongation and thus, body height. It is clearly established that the rates of skeletal growth are determined by chondrocyte proliferation (minor contribution) and chondrocyte hypertrophy (major contribution) in the growth plate. It is also well established that different growth plates within the same organism at any given time "grow" at very different rates, such as proximal and distal growth plates in radius and ulna. None of the links established by the data presented here directly address (or explain) chondrocyte proliferation or hypertrophy and genes known to regulate them positively or negatively (PTHrp, Indian hedgehog, FGFs, IGFs, etc.). The only links established are to general aspects of chondrocyte and skeletal biology, thus making it very difficult to understand how the links have anything to do with "height". The behavior of the growth plate is controlled by intrinsic mechanisms (see above) as well as systemic factors such as growth hormone. Again, none of the links related to this or other systemic cues.

Thank you for raising these points about linking our epigenetic data with biological insights into chondrocyte and skeletal biology. In our revision, we have included several improvements to strengthen the connection between our ATAC-seq data and chondrocyte biology. First, we revised our GREAT analysis by removing ATAC-seq peaks that are also seen in the mouse brain. By doing so, we are limiting ourselves to more chondrocyte-specific regions. When repeating the GREAT analysis, we find many chondrocyte-relevant gene sets, such as “Cartilage condensation”, “Positive regulation of chondrocyte differentiation”, and “Chondrocyte proliferation”. Second, we have included an additional figure (new Figure 1—figure supplement 1) where we show the landscape of ATACseq peaks near several genes with prior connections to chondrocyte biology – *Ihh, Pth1r*, and *Fgfr1*. We believe that these revisions help more directly connect our data to chondrocyte biology.

The authors made use of the proximal and distal portions of E15.5 mouse embryo femur and associated perichondrium as starting material for the epigenetic analysis. The tissues were digested into single cell suspension enzymatically before processing for ATAT-seq. There is no explanation of why this developmental stage was chosen and why both ends of the femur were considered. Also, the lengthy enzymatic digestion could have altered gene expression and chromatin configuration.

Thank you for raising this important point regarding our choice of using E15.5 embryos. We have addressed this point in our response to the Editor’s first comment (see above).

To establish stronger links, the authors considered the most compelling GWAS variants overlapping ATAT-seq peaks, finding 59 variants distributed over 26 loci. The two examples mentioned are CHSY1 and HOXD13 neither of which has established roles in growth plate function. Indeed, most evidence indicates that HOXD13 regulates autopod skeletal patterning and morphogenesis, but not rates of growth.

We explored the roles of a number of GWAS loci in the revised edited manuscript, highlighting ATAC-seq regions that overlap with previously published growth plate enhancers, reside near classic genes involved in growth plate biology, and display evidence of overlapping with human height GWAS variants. We also now have demonstrated evidence of their functional involvement in chondrocyte biology. For this manuscript, we focused on *CHSY1* as one example relevant to both human height and human health, as it has previously published roles in chondrocyte biology and mutations in *CHSY1* cause a variety of skeletal dysplasias (see main text). We identified HOXD13 as a potential transcriptional regulator of *CHSY1* at the rs9920291 enhancer variantusing the UniProbe Database (see Materials and methods). While it is well-known that Hoxd13 plays important roles in autopod patterning, it is also expressed in growth plates (Reno et al., 2008) and has been known for some time (along with other Hox genes) to be growth promoting factor (Morgan and Tabin,’ 94; Yokouchi et al.,’ 95; Goff and Tabin,’ 97; Capecchi,’ 97; Papenbrock et al., 2000; Zhao and Potter, 2001). As Hox proteins act redundantly during limb bud development and growth plate function, deciphering the specific functions of Hoxd13 or its targets has been extremely difficult. However, our newly added data reveal that HOXD13 is expressed in chondrocytes and can increase *CHSY1* expression when overexpressed in chondrocyte cell lines (new Figure 6).

The authors make a strong case that the data can provide insights into the meaning and functional relevance of GWAS correlations, but I do not see how they provide any direct insights into what regulates human height. It may be worth considering an overall change in rationale for the study.

We thank the reviewer for raising this point regarding the framing of our work. However, we believe that our work provides many important insights into the regulation of human height. First, we identified 44 genes at height GWAS loci, where the gene is differentially expressed in the growth plate and where there is a height GWAS variant that overlaps ATAC-seq peaks. These 44 genes represent a subset of height-associated genes that likely regulate height at the growth plate. Second, we have identified several transcription factors with binding sites at these ATAC-seq peaks, providing additional insight into the transcriptional controls underlying height. Finally, as detailed above, we now provide functional experiments detailing how common regulatory variation at *CHSY1* influences height. These results provide novel insights into the genes, regulatory circuitry, and mechanisms underlying control of height at the growth plate.

Reviewer #3:[…] The authors proceed to identify height GWAS loci and specific SNPs that overlap femur open chromatin peaks, and provide potential hypotheses for how select SNPs (e.g. rs9920291) may act to influence human height. Unfortunately, despite claiming the identification of "compelling mechanisms for GWAS variants" (Abstract), the authors do not provide sufficient experimental or computational evidence in support, except for rs4911178 that references data in Capellini et al. 2017 (PMID 28671685). In summary, the authors present an important epigenomic dataset that can act as a first step for future study of height GWAS SNPs, but do not provide "strong evidence" (subsection “Known and novel targets of human height variants in putative chondrocyte regulatory regions”, first paragraph) for compelling mechanisms at any height GWAS loci beyond referencing their previously published paper.

We thank the reviewer for his/her careful review our paper and are pleased that the reviewer recognizes the value of our work. To strengthen our connections between ATAC-seq peaks at GWAS loci with biological mechanisms, we now provide extensive functional work at the *CHSY1* locus. We have also made many other changes and performed new experiments as suggested by the reviewer.

1) Substantially fewer GWAS loci overlap ATAC-seq peaks when using the fine-mapped 95% credible variants, instead of all SNPs with r^2^>0.5 (e.g. 192 loci vs. 317 loci, subsection “Fine-mapped height variants overlap with femoral growth plate open chromatin regions” and subsection “Human height variants are enriched in femoral open chromatin regions”, last paragraph; also 26 loci vs. 46 loci, subsection “Known and novel targets of human height variants in putative chondrocyte regulatory regions”, first paragraph and subsection “A subset of human height variants in femoral open chromatin regions reside near genes differentially expressed in the growth plate”, second paragraph). Given that r^2^>0.5 is not a strict LD threshold (e.g. Wood et al. 2014 uses r^2^>0.8 for strict LD), is it still meaningful to report overlap statistics using the r^2^>0.5 set when the PICS 95% credible set is available?

We thank the reviewer for raising these important points regarding proxy SNPs at GWAS loci. We agree with the reviewer that r^2^>0.5 is not a universally agreed upon LD threshold. We have now included GoShifter analyses at several different r^2^thresholds ranging from 0.2 to 0.8 (new Figure 3—figure supplement 1). Among the various other LD thresholds tested, an r^2^ cutoff of 0.5 demonstrated the strongest enrichment (p=0.0059).

The reviewer makes an excellent point regarding whether we should just report PICS credible set overlaps. While the reviewer’s point is valid and well-taken, we have decided to be more inclusive in our approach and report both PICS overlaps and variants overlaps with r^2^>0.5 in our new Supplementary file 6. This choice was made particularly in light of potential issues with using PICS, as PICS may not be the best fine-mapping tool. Additionally, the height GWAS was imputed to HapMap (as compared to a more dense panel such as 1000 Genomes), limiting our ability to properly fine-map. We do wish to note that we were limited to using PICS for fine-mapping, as to our knowledge, PICS is the only fine-mapping software that doesn’t require densely imputed summary statistics, but only requires a set of lead variants and LD relationships.

2) The authors state "44 different genes have been identified as possibly being modulated by these putative regulatory variants". However, in the present manuscript the authors do not provide experimental or computational evidence implicating the ATAC-seq peaks in regulating the nearby growth plate genes, and provide limited evidence that any of these overlapping variants can alter transcriptional regulatory activity.

Thank you for this comment. We now provide evidence that differentially expressed growth plate genes that are also eQTLs genes in whole blood are enriched at ATAC-seq peaks (see response to point #6). Additionally, as detailed in our response to the Editor’s third comment, we now provide extensive experimental evidence for rs9920291 as a causal variant at the *CHSY1* locus.

3) Following point #2, and to support the authors' claim of identifying "prime targets for functional testing", it would be helpful to know how many GWAS SNP overlaps with femur open chromatin regions the authors believe to be biologically meaningful versus coincidence.

We thank the reviewer for raising this important point. Unfortunately, without a “truth” set, it is difficult to evaluate what proportion of our nominated SNPs are likely to be biologically meaningful as opposed to coincidence. However, we do note that using both GoShifter and random matched GWAS loci, we detect very strong enrichments of our height GWAS SNPs at ATAC-seq loci, suggesting that a large proportion of our nominated SNPs are meaningful. We also point out the work of Ulirsch et al., Cell, 2016, which performed MPRA for red blood cell traits and demonstrated that a large proportion of variants nominated by similar approaches as used in our paper demonstrated functional activity in the MPRA assay. As an additional approach to demonstrate that our observed overlaps are not simply coincidence, we also performed GoShifter analyses for enrichments of GWAS associations of several traits (type 2 diabetes, total cholesterol, body mass index, schizophrenia, coronary artery disease, and age of menarche) at our ATAC-seq peaks. For most of these additional traits, we did not detect significant associations (p>0.05) of enrichment (new Figure 3—figure supplement 1). Lastly, as described elsewhere, we now provide new data at the *CHSY1* locus to demonstrate the value of our approach.

4) Similar to point #2, it is premature to state "59 variants thus have strong evidence for being potentially causal variants". For the authors' example of the CHSY1 locus, while there are 4 SNPs in the PICS 95% set that overlap an ATAC-seq peak nearby CHSY1, Supplementary file 6 reveals there are 87 SNPs in the 95% credible set at the CHSY1 locus. Without more evidence, it is difficult to agree with the potential causal variant conclusion.

We recognize that it is difficult to determine what proportion of our prioritized variants will turn out to be causal. However, we believe that our ATAC-seq data is bolstering the evidence of these variants being causal. For example, at *CHSY1*, intersection with ATAC-seq data prioritized the 87 SNPs in the 95% credible set down to just four variants. In our revision, we now provide extensive functional evidence for rs9920291, providing further credence for our claim that our approach can nominate likely causal variants (new Figure 6).

5) Subsection “Known and novel targets of human height variants in putative chondrocyte regulatory regions”, first paragraph: The authors suggest rs9920291 disrupts HOXD13 binding, but do not provide evidence that HOXD13 actually binds this site at the endogenous locus. As computationally predicted TF motifs are prevalent in the genome, it is hard to assess whether this SNP/motif overlap is biologically meaningful.

We agree with the reviewer that predicted motifs are prevalent throughout the genome. While the HOXD13 binding prediction is based on experimental HOXD13 DNA-binding specificities (Hume et al. 2015), there are limited ChIP-seq datasets in relevant tissues to assess whether HOXD13 truly binds at rs9920291, in part because of the known difficulty of generating and using HOXD13-specific antibodies for ChIP-seq. We have toned down the language in the manuscript regarding this HOXD13 connection (subsection “Known and novel targets of human height variants in putative chondrocyte regulatory regions”, second paragraph). Additionally, we do now provide evidence that overexpression of HOXD13 increases expression of *CHSY1* (new Figure 6).

6) Wood et al. (2014) found that height GWAS loci were more likely to be eQTLs (in blood) for genes expressed in cartilage. Did the authors examine whether the 468 credible SNPs overlapping femur open chromatin peaks are enriched for eQTL signals? This analysis may strengthen the authors' mechanistic claims.

We thank the reviewer for this excellent suggestion. We have now included additional analyses, which confirm the observation in Wood et al., that height GWAS loci are more likely to be eQTLs in blood for genes expressed in cartilage. To do this, we used a set of eQTLs from whole blood from Westra et al., 2014 and subsetted for eQTLs genes that are differentially expressed in the growth plate. We then ran GoShifter on this set of corresponding eQTLs variants and indeed detected a significant enrichment when compared with our ATAC-seq peaks (p=0.0004). This is discussed in the third paragraph of the subsection “A subset of human height variants in femoral open chromatin regions reside near genes 261 differentially expressed in the growth plate”.